# LESS DATA, FASTER TRAINING: REPEATING SMALLER DATASETS SPEEDS UP LEARNING VIA SAMPLING BIASES

**Jingwen Liu**
Columbia University
jingwenliu@cs.columbia.edu

**Ezra Edelman**
University of Pennsylvania
ezrzae@cis.upenn.edu

**Surbhi Goel**
University of Pennsylvania
surbhig@cis.upenn.edu

**Bingbin Liu**
Kempner Institute, Harvard University
bliu@g.harvard.edu

## ABSTRACT

This work investigates the "small-vs-large gap", where repeating on *fewer samples* can lead to *compute saving* during training compared to using a larger dataset. This is observed across algorithmic tasks, architectures and optimizers and cannot be explained using prior theory. We argue that the speedup comes from appropriate layer-wise growth enabled by *sampling biases*, which is more pronounced when the dataset size is smaller. We provide both theoretical analysis and empirical evidence from various interventions. Our results show that using a smaller dataset with more repetitions is not just a fallback strategy under data scarcity, but can be proactively leveraged as a favorable inductive biases for optimization, particularly in reasoning tasks.

## 1 INTRODUCTION

Data plays a central role in machine learning. The conventional wisdom is that more data is better, a view supported by both classic generalization theory and extensive empirical evidence (Muennighoff et al., 2023). Surprisingly, recent work has revealed a counterintuitive phenomenon: under a fixed compute budget, training on a smaller dataset can outperform training on a larger set. This has been observed on the well-studied single-index models (Dandi et al., 2024; Arnaboldi et al., 2024; Lee et al., 2025) and various algorithmic tasks (Charton and Kempe, 2024).

We refer to this phenomenon as the *small-vs-large (dataset) gap*. We find this exists generally across various tasks for both MLP (Figure 1) and Transformers (Figure 9), including settings where prior theory fail to explain. Instead, we show that the small-vs-large gap primarily results from favorable optimization biases due to **sampling biases** of the dataset.

- We confirm that the small-vs-large gap exists across various settings. Given a fixed task, architecture, and optimizer, the gap is observed in terms of both the number of *optimization steps* and the overall *compute complexity*, which depends on both the number of steps and the per-step cost which is proportional to the batch size.

- We show that *sampling biases* induced by smaller datasets is a primary driver of the small-vs-large gap (Section 3): sampling biases modulate the relative magnitude of updates across layers, which in turn helps with feature learning. We identify regimes where existing theory [1] fails to explain the gap (Appendix A), and theoretically show that training on smaller datasets reduces the number of steps for MLP convergence (Theorem 1).

- We further support the theoretical explanation with a broad set of empirical evidence (Section 4). First, training on a small dataset with *random labels* leads to a speedup comparable to that observed with real labels, indicating that sampling bias is the main

---

[1]Appendix .1 provides more discussion on the related work.

(a) SGD on $(20, 6)$-parity  (b) GD on $(20, 6)$-parity.  (c) SGD on SIM, $d = 40$.  (d) GD on SIM, $d = 40$.

Figure 1: **Small-vs-large gap exists** for both mini-batch (SGD) and full-batch (GD) training. The gap under GD is a notable example that prior theory fails to capture. Results are based on SIM and parity on 2-layer MLPs. Transformer results on more tasks are similar and deferred to Appendix C.3.

> mechanism, as the gap persists without task-relevant signal. Moreover, *parameter-wise interventions* substantially reduce the small-vs-large gap, including adjustments to initialization scales and parameter-wise learning rates across both MLP and Transformers. For Transformers, our findings additionally suggest that the widely used QK normalization has nuanced optimization effects that merit further investigation.

Together, our results suggest that training on a smaller dataset with increased repetitions is not merely a fallback under data scarcity, but a source of beneficial optimization inductive biases that can be leveraged more proactively, particularly for reasoning tasks.

## 2 LESS DATA CAN LEAD TO FASTER LEARNING

We present empirical evidence that less data leads to accelerated learning across setups. We focus on synthetic tasks, which have tunable parameters and thus allow for explicit control over task complexity. We start with two classic feature learning tasks which have been extensively studied in the literature.

- **Single-index model (SIM)**: the input is a Gaussian vector $x \sim \mathcal{N}(0, I_d)$, and the label is $y := \phi(\langle w^*, x \rangle)$, where $w^*$ is the ground truth feature vector, and $\phi : \mathbb{R} \to \mathbb{R}$ is an unknown link function. Our experiments take $\phi$ to be a Hermite polynomial.

- $(d, k)$**-sparse parity**: the input is a boolean vector $x \sim \text{Unif}(\{\pm 1\}^d)$, and the label is given by $y := \prod_{i \in S} x_i$, where $S \subset [d]$ is an unknown support of size $k$.

We compare performance under a given compute budget, as measured by the batch size × number of optimization steps. Results are averaged over 32 random seeds unless otherwise specified. Details on data use and experimental setups are deferred to Appendix C.

**Mini-batch updates**  We start with mini-batch training, where the batches are sampled uniformly over a training set. This is the common training strategy in practice and where prior work has also reported the small-vs-large gap (Charton and Kempe, 2024). As shown in Figure 9, smaller datasets lead to faster convergence for all tasks. For SIM, in-context learning regression and modular addition, multi-phase is used to balance accelerated optimization and good generalization.

However, mini-batch updates introduce a confounding factor of the *number of repetitions*: when trained for the same number of steps, each sample in a smaller dataset is reused more frequently over the course of training. It is thus unclear whether this increased repetition is the primary source of the observed speedup. Our subsequent gradient descent results show that this is not the case.

**Gradient descent (full-batch updates)**  We sample datasets of varying sizes from the population and run (full-batch) gradient descent on each dataset. As shown in Figure 1, smaller datasets have better performance at each time step throughout training. Moreover, the total saving in compute is more significant than the reduction in steps, since smaller datasets incur lower per-step computational cost. For instance, in Figure 1(b), using $N = 2^{14}$ con-

verges in 1500 steps whereas the full population (i.e. $N = 2^{20}$) [2] requires more than 2000 steps, leading to a 100x speedup in compute.

**Other setups** We provide similar results on Transformers across more tasks in Appendix C.3. To understand the implications under scaling, we show that increasing the model depth or the task complexity both amplify the small-to-large gap; details are provided in Appendix D, along with discussions on when we do *not* expect the gap to hold.

## 3 UNPACKING THE EFFICIENCY GAIN FROM SMALLER DATASETS

We use sparse parity as a sandbox for understanding the small-vs-large gap. Notably, alternative theories in existing work are not sufficient to explain the gap. The observed speedup in our GD experiments renders any SGD-based explanations invalid, including work on CSQ-SQ gap (Dandi et al., 2024; Arnaboldi et al., 2024; Lee et al., 2025), gradient variances (Johnson and Zhang, 2013; Kotha et al., 2025). Closer to our work are explanations based on biased input distributions (Valiant, 2012; Abbe et al., 2023; Kalai et al., 2009; Cornacchia et al., 2025); however, the effect of bias in their analysis is exponentially small in the sparsity $k$, whereas we leverage the sampling bias which is independent of $k$ and depends only on the dataset size. We defer the details to Appendix A.

### 3.1 OUR EXPLANATION: DATASET SAMPLING BIAS ACCELERATES THE RELATIVE NORM GROWTH BETWEEN LAYERS

We instead argue that a primary source of the speedup from smaller datasets is that their sampling biases implicitly adjust the norm scale across layers, effectively changing the per-layer learning rates. Intuitively, for parity learned with 2-layer MLPs, feature learning occurs in the first (i.e., input) layer. The first layer learning speeds depends on the second layer, whose growth is faster when the dataset is smaller, due to stronger biases. The rest of this section formalizes this intuition, and we provide empirical evidence in Section 4.

The theoretical analysis considers a 2-layer network with quadratic activation, i.e. $f(x) = a\sigma(w^\top x) - 1$, where $\sigma(z) := \frac{1}{2}z^2$, on the 2-sparse parity task. The model is optimized using the correlation loss, i.e. $L(f) = \mathbb{E}_{x,y}[\ell(y, f(x))]$ where $\ell(y, y') = -yy'$, with learning rate $\eta$ and projected updates, i.e., $a^{(t)} = \max\{-1, \min\{1, a^{(t)}\}\}, w^{(t)} = \frac{w^{(t)}}{\|w^{(t)}\|_2}$ at each step.

Following standard practices, we initialize with $w^{(0)} \sim \text{Unif}(\mathbb{S}^{d-1})$, and $a^{(0)} \sim \mathcal{N}(0, 1/m)$ where $m$ can be considered as a model width parameter. Since the correlation loss does not introduce interaction across neurons, the analysis below can be considered as focusing on one neuron of a width-$m$ network. We show that a smaller $N$ leads to faster convergence:

**Theorem 1** (2-phase training from standard initialization). *Consider a 2-phase training with $m > d$ and learning rate $\eta = \Theta(1)$. The first phase uses a randomly sampled dataset of size $d \leq N \leq d^2$, until $|a| \geq a_\star$ for some $0 < a_\star \lesssim \frac{1}{(Nd)^{1/4}(\log d/\delta)^{1/2}}$; the second phase uses the full population gradient, until reaching a $\hat{w}$ such that $\|\hat{w} - w^\star\|_2 \lesssim \sqrt{\varepsilon}$. Let $T_1, T_2$ denote the numbers of steps required in each phase respectively. Let $p_{\text{all}} \in (0, 1)$ be a universal constant where $p_{\text{all}} = \Theta(1)$. [3] Then, with probability at least $p_{\text{all}} - \delta$,*

$$T_1 \lesssim \frac{a^* \sqrt{N}}{\eta}, \quad T_2 \lesssim \frac{2}{\eta a^*} \log\left(\frac{d}{\varepsilon}\right). \tag{1}$$

*With the optimal choice of $a_\star$, the total number of steps is $O\left((Nd)^{1/4} \log\left(\frac{d}{\varepsilon}\right) \log^{1/2}\left(\frac{d}{\delta}\right)\right)$.*

Theorem 1 implies that a small $N$ leads to a direct saving in optimization steps. One could alternatively skip Phase 1 and train directly on the full population, which would require $O(m^{1/2} \log(d/\epsilon))$ steps (Lemma 6), worse than the 2-phase convergence when $m \gg d^2$.

---

[2] Although the population gradient for sparse parity reveals information about the support (Barak et al., 2022), it is not sufficient to enable fast convergence in practice; see Remark 5 for details.

[3] $p_{\text{all}}$ is formally defined in Lemma 10.

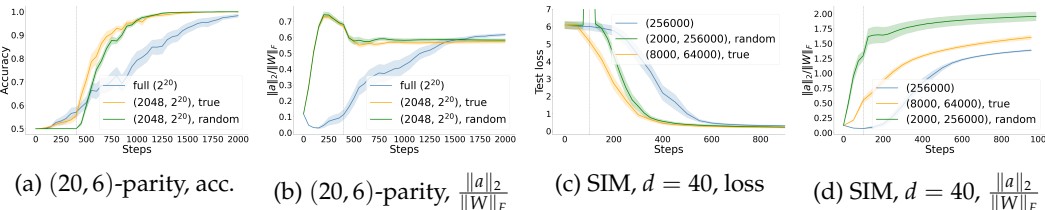

(a) $(20, 6)$-parity, acc.    (b) $(20, 6)$-parity, $\frac{\|a\|_2}{\|W\|_F}$    (c) SIM, $d = 40$, loss    (d) SIM, $d = 40$, $\frac{\|a\|_2}{\|W\|_F}$

Figure 2: **Training on small datasets with *random labels* leads to faster learning**. Shown for GD, the initial random-training leads to speedup and faster growth of $\|a\|_2 / \|W\|_F$. The blue/yellow curves correspond to large/small sets. The green curves correspond to training first on a small set of *random* labels and then switching to large sets with true labels.

## 4 EMPIRICAL EVIDENCE FOR THE EFFECTIVE LEARNING RATE HYPOTHESIS

This section provides empirical evidence supporting the claim in Section 3 that the small-vs-large gap is due to the relative growth rates of the two layers. Figure 7 provides direct observational evidence: during the initial phase of training, the weight norm ratio $\|a\|_2 / \|W\|_F$ increases more rapidly when the dataset is smaller, for both parity and SIM.

We further consider *interventions* on the training process to provide stronger evidence, in terms of 1) *data*, where we show that the speedup from small-set exists even when the labels are random; 2) *weight norms*, by changing initialization scale or the adoption of normalization layers; and 3) layer-wise *learning rate* controls.

**Training first on *random* labels also leads to faster learning**    The phase-1 analysis for Theorem 1 extends to datasets with random labels, which have similar sampling biases and hence should achieve a similar speedup. Our empirical results confirm this: speedup exists even when training first on a small set of *random* labels and then switching to the large-set training (see Appendix C.4 for details). Figure 2 shows MLP results on parity and SIM. Both the accuracy and the weight norm growth (measured by $\|a\|_2 / \|W\|_F$) are sensitive only to the dataset size, but not the label choice (i.e., true or random labels). This also agrees with our theory (i.e., phase 1 analysis) that the benefit of sampling bias exists even with random labels.

**Weight norm scaling**    Theorem 1 implies that directly increasing the second layer should lead to accelerated learning. For MLP, we control the weight norms by altering the **initialization scales**, where the variance used for weight initialization are multiplied with layer-specific constants. Figure 3 compares the test accuracies of training on full population versus a random subset, across initialization scales. While the full population performs worse than the subset at the default initialization [4] (marked by the red star), there exist scaling constants that completely close the small-vs-large gap. However, identifying the right constants requires extensive searching, whereas training on random subsets is able to adjust the scaling automatically and is robust to initialization scale.

We note that such scaling change is different from the $\mu P$ parameterization (Yang and Hu, 2020; Yang et al., 2022), and we discuss the comparison in Appendix C.5.1. Similar results extend to Transformers, where we consider the scaling introduced by the QK normalization and layers of the per-block MLP; results are deferred to Appendix C.5.2.

**Layer-wise learning rates**    For MLP with ReLU activations, growing one layer leads to larger updates on another. Hence the layer-wise norm adjustment can be equivalently achieved by adjusting layer-wise learning rates. Let $\eta_1, \eta_2$ denote the learning rates for the first and second layer, respectively. Figure 4 shows that the optimal choice of $(\eta_1, \eta_2)$, where $\eta_1 \gg \eta_2$, significantly reduces the small-vs-large gap between the full population ($N = 2^{20}$) versus a random subset ($N = 2^{14}$).

---

[4]We use the PyTorch default, i.e., initializing a linear layer $W$ with $W_{ij} \sim \text{Unif}[-1/\sqrt{d_{\text{in}}}, 1/\sqrt{d_{\text{in}}}]$.

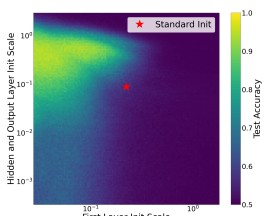
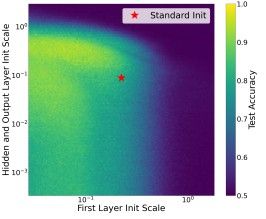

(a) Full population (size $2^{20}$)   (b) Random subsets of size $2^{14}$

Figure 3: **Smaller-set training is more robust to initialization scale.** $(20, 6)$-parity on two-hidden-layer MLPs after 1024 SGD iterations. Each pixel corresponds to the learning rate with the best accuracy, averaged over 256 random seeds.

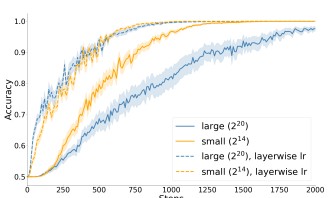

Figure 4: **Layer-wise learning rate shrinks the full-vs-sub population gap.** Results are shown for MLP on $(20, 6)$-parity, trained with GD.

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

CONTENTS

## .1 RELATED WORK

It is widely believed that in deep learning, more is better, as captured by the study of scaling laws. However, different resources may not need to be scaled together. For instance, *data repetition*, which keeps the sample size fixed while scaling up compute, can achieve similar performance to compute-matched online training when the amount of repetition is moderate (Xu et al., 2021; Sekhari et al., 2021; Muennighoff et al., 2023; Lin et al., 2025; Yan et al., 2025). We are interested in the more extreme phenomenon termed the *small-vs-large gap*, where *reducing* the sample size when holding compute constant can help improve performance. The small-vs-large gap has been observed in recent work on algorithmic tasks (Charton and Kempe, 2024), in-context learning (Zucchet et al., 2025), and language model finetuning on reasoning tasks (Kopiczko et al., 2026). Prior work has shown this for learning single-index models, where taking more than one gradient steps on the same set of samples can reduce the total number of gradient steps (Dandi et al., 2024; Arnaboldi et al., 2024; Lee et al., 2025). The intuition is that while online SGD lies in the class of correlational statistical query (CSQ) algorithms, SGD with sample repeats belongs the more general class of statistical query (SQ) algorithms. In contrast, we find that the compute savings from using less data hold even in regimes where the CSQ-SQ distinction does not apply, including training with full-batch gradient descent, and tasks with discrete domains. Close to our quadratic setting in Section 3.1, a concurrent work by Kovačević et al. (2026) shows the statistical advantage of full-batch gradient descent over SGD where mini-batches are sampled fresh from the population. A key difference is that the model in Kovačević et al. (2026) only has a single layer (i.e., $f(x) = \sigma(w^\top x)$), hence the effect of relative weight norm does not apply.

Previous work has also studied how multi-pass SGD can improve the sample complexity over single-pass SGD in various settings, including linear regression (Pillaud-Vivien et al., 2018; Lin et al., 2025), general stochastic convex optimization (Sekhari et al., 2021), and non-convex problems under the PL condition (Xu et al., 2021). A crucial difference from our work is that these results focus on saving *samples* but not the *compute*: they show that the population error achieved by $T$ online SGD steps, where each step is taken on an iid sample from the population, can be achieved by $T$ steps of multi-pass SGD, where each step is taken on an iid sample drawn from an empirical distribution of size smaller than $T$. In contrast, we will show that it is possible to achieve the same error as $T$ online SGD steps using *fewer than T steps* of multi-pass SGD.

We will show that the key mechanism behind such speedup comes from the strong sampling biases from smaller datasets, which effectively adjusts the relative update speeds across the layers, leading to faster learning. Such adjustments relate to the idea of balancing contributions from different layers, which has been studied extensively in the optimization and feature learning (Yang and Hu, 2020; Azulay et al., 2021; Yang et al., 2022; 2023; Everett et al., 2024).

## A EXTENDED DISCUSSION ON PRIOR THEORY

**SQ-CSQ difference** One mechanism of acceleration identified by prior work is that taking multiple gradient updates on the same data effectively transform (batch) SGD from a correlational statistical query (CSQ) algorithm to a statistical query (SQ) algorithm (Dandi et al., 2024; Arnaboldi et al., 2024; Lee et al., 2025), with the latter being a more powerful class of algorithms. Prior work uses this to explain the speedup for batch SGD on single-index model (SIM), for which there is a known gap between CSQ and SQ lower bounds.

While insightful, the CSQ-SQ gap cannot fully explain the observed speedup. It cannot explain why there is a speedup for SIM even when using (full-batch) GD, where smaller and larger datasets are both repeated and for the same amount of times. Moreover, it is not applicable to the wide range of tasks where the SQ and CSQ lower bound coincide, which include all discrete problems such as sparse parity and mod addition.

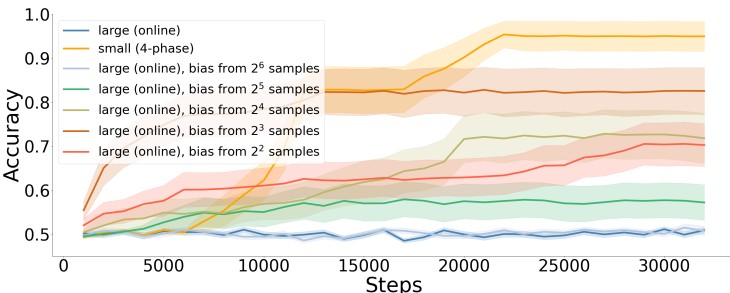

Figure 5: **Biasing online training does not bridge the speed gap.** For sparse parity ($d = 20, k = 6$), biasing the Bernoulli distribution with the empirical mean of $2^i$ samples (for $i \in \{2, 3, 4, 5, 6\}$) makes online training faster for certain values of $i$ (best at $i = 3$). However, to reach similar speedup as given by training on smaller datasets (marked as "4-phase"), the amount of bias required for large set (i.e., online) training would require an extremely small dataset size.

**Gradient variance reduction**   It is known that reducing gradient variances help speed up convergence (Johnson and Zhang, 2013), and (Kotha et al., 2025) has investigated that gradient variance reduction interventions can accelerate training in sparse parity and in-context linear regression tasks. However, this cannot explain the speedup with gradient descents, where each step takes the full dataset and has no sampling-induced variances.

**Biased (input) distribution**   Specific to sparse parity and SIM, prior theory indicates another explanation from the benefit of biased distributions. The focus was on the biases in the input distribution, which can either be explicitly constructed sparsity (Valiant, 2012; Abbe et al., 2023), or randomly perturbed distributions (Kalai et al., 2009; Cornacchia et al., 2025). Our setup does not select for sparsity and is closer to the latter. However, a critical difference is that the signal strength (in terms of first-order Fourier coefficients) in Cornacchia et al. (2025) is exponential in the sparse $k$, whereas the sampling bias in our analysis depends on the dataset bias only and is independent of the sparsity. In particular, for a random subset of size $N$, the signal strength in Cornacchia et al. (2025) is $O(N^{-k/2})$, which is much smaller than the $O(N^{-1/2})$ sampling bias.

However, these biases are not sufficient to explain the speedup. We provide two types of empirical evidence: that online training with biased distribution does not lead to the same amount of speedup, and that the speedup persists even when the dataset bias is removed.

Online training with biased distributions has minor effects. We train with online data following the biases of an offline dataset. Specifically, for $d = 20$, $k = 6$, we take the biases from $2^i$ samples for $i \in \{4, 6, 8, 10, 12\}$; recall that $2^{14}$ is the size of the dataset used training for $(20, 6)$-parity. As shown in Figure 5, biasing the distribution does improve over training with an unbiased distribution as prior theory predicts. However, unless the samples size is exceedingly small (e.g. fewer than $2^5 = 32$ samples), the speedup is much less compared to training on a small subset.

Small datasets without biases still lead to speedup. Recall from earlier that at initialization, the model is close to a saddle point of the population loss. Building on this, we experiment on parity with inputs where $\mathbb{E}[x] = 0$, $\mathbb{E}[y] = 0$, and $\mathbb{E}[x|y = -1] = \mathbb{E}[x|y = 1] = 0$ for parity. This means matching all 1st-order statistics of the empirical dataset to those of the population, which will lead to a 0 gradient at $w = 0$ in the single-neuron example. We find similar results on SIM, where the inputs are whitened to match both 1st and 2nd order statistics, i.e. transform the dataset with $\tilde{x} = \hat{\Sigma}^{-1}(x - \hat{\mu})$, where $\hat{\mu}$ and $\hat{\Sigma}$ are the empirical mean and covariance. For both tasks, the speedup provided by smaller datasets persists (Figure 6).

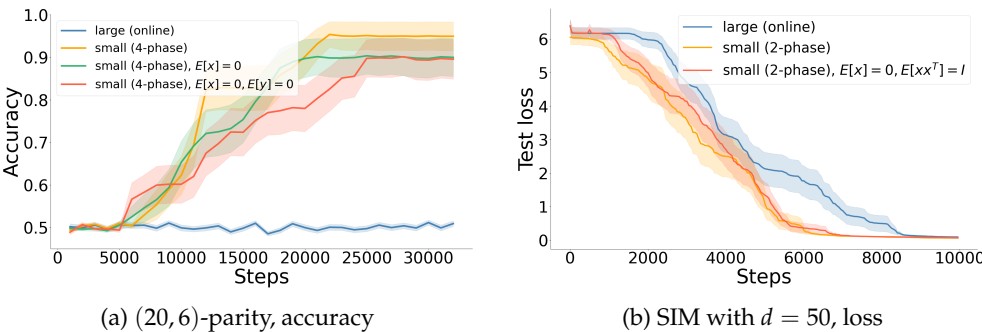

(a) $(20, 6)$-parity, accuracy         (b) SIM with $d = 50$, loss

Figure 6: **Repetition remains superior with dataset bias removed**. Results are based on mini-batch updates. Parity experiments are with Transformers, and SIM experiments are with MLP.

## B  PROOF DETAILS

Motivated by the empirical evidence in Sections 2 to 4, we analyze a minimal setting, a single quadratic neuron on 2-sparse parity under a two-phase schedule: phase 1 runs GD on a fixed dataset of size $N$, and phase 2 switches to the full population.

### B.1  SETUP

We assume the input $x \in \{\pm 1\}^d$ is sampled uniformly from the hypercube and the label $y = x_1 x_2 \in \{\pm 1\}$ is a 2-sparse parity. We study the quadratic neuron

$$f(x) = \frac{1}{2} a (w^\top x)^2,$$

trained with correlation loss $\ell(y, \hat{y}) = -y\hat{y}$. Let $w^\star$ denote a global minimizer, where $|w_1^\star| = |w_2^\star| = \frac{1}{\sqrt{2}}$, and $w_i = 0$ otherwise.

**Projection.** To ensure that the weights are bounded at the solution, we use the following projections:

- **Output weight clipping:** after each update we set $a \leftarrow \text{clip}_{[-1,1]}(a)$.

- **Input weight renormalization:** after each update we set $w \leftarrow w / \|w\|_2$.

We consider 2-phase training, where Phase 1 uses a randomly sampled dataset of size $N$, and Phase 2 uses the full population.

**Phase 1 (fixed batch of size $N$).** Fix a dataset $\{(x^{(s)}, y^{(s)})\}_{s=1}^N$ and define the empirical moment matrix

$$\widehat{M} := \widehat{\mathbb{E}}[yxx^\top] := \frac{1}{N} \sum_{s=1}^N y^{(s)} x^{(s)} x^{(s)\top}.$$

One step of (projected) gradient descent on this fixed batch takes the form

$$a^{(t+1)} = \text{clip}_{[-1,1]}\left( a^{(t)} + \frac{\eta}{2} (w^{(t)})^\top \widehat{M} w^{(t)} \right), \tag{B.2}$$

$$w^{(t+1)} = \frac{w^{(t)} + \eta a^{(t)} \widehat{M} w^{(t)}}{\left\| w^{(t)} + \eta a^{(t)} \widehat{M} w^{(t)} \right\|_2}. \tag{B.3}$$

Let's define

$$q^{(t)} := (w^{(t)})^\top \widehat{M} w^{(t)}$$

which will be useful in the analysis.

**Phase 2 (population).** After phase 1, we switch to population gradients, replacing $\widehat{M}$ by the population matrix

$$M := \mathbb{E}[yxx^\top] = e_1 e_2^\top + e_2 e_1^\top.$$

in the above updates.

We restate Theorem 1 below, which shows that using a smaller $N$ in Phase 1 improves convergence.

**Theorem** (2-phase training from standard initialization; Theorem 1 restated.)**.** *Consider a 2-phase training with $m > d \geq 3$ [5] and learning rate $\eta \leq \frac{1}{2}$. The first phase uses a randomly sampled dataset of size $d \leq N \leq d^2$, until $|a| \geq a_\star$ for some $a_\star \in (0,1)$ where $a_\star \lesssim \frac{1}{(Nd)^{1/4}\sqrt{\log(d/\delta)}}$; the second phase uses the full population gradient, until reaching a $\hat{w}$ such that $\|\hat{w} - w^\star\|_2 \lesssim \sqrt{\varepsilon}$. Let $T_1, T_2$ denote the numbers of steps required in each phase respectively. Let $p_{\text{all}} \in (0,1)$ be a universal constant where $p_{\text{all}} = \Theta(1)$. [6] Then, with probability at least $p_{\text{all}} - \delta$ over the random initialization and the phase-1 samples,*

$$T_1 \lesssim \frac{a_\star \sqrt{N}}{\eta}, \quad T_2 \lesssim \frac{2}{\eta a_\star} \log\left(\frac{d}{\varepsilon}\right). \tag{B.4}$$

*On the same event, with the optimal choice of $a_\star$, the total number of steps is $O\left((Nd)^{1/4} \log\left(\frac{d}{\varepsilon}\right) \log^{1/2}\left(\frac{d}{\delta}\right)\right)$.*

## B.2 ANALYSIS OF PHASE 1

First we show that a smaller dataset size $N$ leads to faster growth of the outer weight.

**Theorem 2** (Phase 1 is faster with fewer fixed-batch samples)**.** *Initialize $w^{(0)} \sim Unif(\mathbb{S}^{d-1})$, and $a^{(0)} \sim \mathcal{N}(0, 1/m)$ where $m$ can be considered as a model width parameter. Fix any target $a_\star \in (0,1]$ and learning rate $\eta > 0$. Consider phase 1 training on a fixed batch of size $N$ and let*

$$T_\star(N) := \min\{t : |a^{(t)}| \geq a_\star\}.$$

*For some choice of $c_0 \in (0, \frac{\sqrt{3}}{2})$, define the event*

$$\mathcal{G}_0 := \left\{ \text{sign}(a^{(0)}) = \text{sign}(q^{(0)}) \text{ and } |q^{(0)}| \geq c_0/\sqrt{N} \right\},$$

*and $p_{\text{good}} := \frac{1}{2} p_{PZ}(c_0)$ where $p_{PZ}(c_0)$ is the constant from Lemma 1. Then $\mathbf{Pr}[\mathcal{G}_0] \geq p_{\text{good}}$. On the intersection of $\mathcal{G}_0$ with the stability event $\{\eta a_\star \|\widehat{M}\|_2 \leq 1/2\}$,*

$$T_\star(N) \leq \left\lceil \frac{2(a_\star - |a^{(0)}|)_+}{\eta c_0} \sqrt{N} \right\rceil = O\left(\frac{a_\star}{\eta} \sqrt{N}\right). \tag{B.5}$$

*In particular, for every $\delta \in (0,1)$, if $a_\star \leq \min\left\{1, \frac{1}{2\eta B_{N,d,\delta}}\right\}$ with $B_{N,d,\delta}$ from Corollary 3, then equation B.5 holds with probability at least $p_{\text{good}} - \delta$. In particular, holding $(\eta, a_\star)$ fixed, the required number of phase 1 steps scales as $\sqrt{N}$.*

*Proof sketch.* The proof consists of three parts.

1. **Initialization gives a nontrivial $q^{(0)}$ at constant probability.** For $w^{(0)}$ randomly sampled from the unit sphere and an i.i.d. batch of size $N$, the empirical quadratic form $q^{(0)} = (w^{(0)})^\top \widehat{M} w^{(0)} = \frac{1}{N} \sum_{s=1}^N y^{(s)} (x^{(s)\top} w^{(0)})^2$ has magnitude $\Omega(1/\sqrt{N})$ with constant probability (Lemma 1). Since $a^{(0)}$ is initialized symmetric about 0 and independent of $q^{(0)}$, we also have $\mathbf{Pr}[\text{sign}(a^{(0)}) = \text{sign}(q^{(0)})] = 1/2$ conditional on $q^{(0)} \neq 0$ (Lemma 2). Together this yields the constant-probability "good" event $\mathcal{G}_0 = \{\text{sign}(a^{(0)}) = \text{sign}(q^{(0)}), |q^{(0)}| \geq c_0/\sqrt{N}\}$.

---

[5] $d \geq 3$ is required for the proof of Lemma 10 regarding the probability of the Beta distribution.
[6] $p_{\text{all}}$ is formally defined in Lemma 10.

2. **While $|a^{(t)}| \leq a_\star$, the sign of $q^{(t)}$ is stable and $|q^{(t)}|$ does not decrease.** Under the stability condition $\eta a_\star \|\widehat{M}\|_2 \leq 1/2$, the normalized updates to the inner weights is a signed power iteration. Lemma 3 shows that the one-step increment $q^{(t+1)} - q^{(t)}$ has the same sign as $a^{(t)}$. On $\mathcal{G}_0$, the outer weight update (with clipping being inactive since $|a^{(t)}|$ hasn't grown to $a_\star < 1$) is $a^{(t+1)} = a^{(t)} + \frac{\eta}{2}q^{(t)}$, hence $\mathrm{sign}(a^{(t)})$ cannot flip as long as $\mathrm{sign}(a^{(t)}) = \mathrm{sign}(q^{(t)})$. Combining these gives by induction that for all $t < T_\star$, $\mathrm{sign}(a^{(t)}) = \mathrm{sign}(q^{(t)}) = \mathrm{sign}(q^{(0)})$ and therefore $|q^{(t)}| \geq |q^{(0)}|$.

3. **Linear growth of $|a^{(t)}|$ and the $\sqrt{N}$ time scale.** On the same event, for all $t < T_\star$ we have $|a^{(t+1)}| = |a^{(t)}| + \frac{\eta}{2}|q^{(t)}| \geq |a^{(t)}| + \frac{\eta}{2}|q^{(0)}|$, so $|a^{(t)}|$ grows at least linearly until it reaches $a_\star$. Thus

$$T_\star \leq \left\lceil \frac{2(a_\star - |a^{(0)}|)_+}{\eta \, |q^{(0)}|} \right\rceil \leq \left\lceil \frac{2(a_\star - |a^{(0)}|)_+}{\eta \, c_0} \sqrt{N} \right\rceil$$

on $\mathcal{G}_0$, which is the claimed $O(\sqrt{N})$ bound.

Finally, Lemma 5 provides a high-probability bound on $\|\widehat{M}\|_2$, yielding an explicit stable choice of $a_\star$ (Corollary 3). □

### B.2.1 CONSTANT-PROBABILITY LOWER BOUND ON $|q^{(0)}| = \Omega(1/\sqrt{N})$

We now prove that, in the parity setting, the fixed-batch quadratic form at initialization

$$q^{(0)} = (w^{(0)})^\top \widehat{M} w^{(0)} = \frac{1}{N} \sum_{s=1}^{N} y^{(s)} (x^{(s)\top} w^{(0)})^2$$

typically has magnitude $\Omega(1/\sqrt{N})$ with *constant* probability.

**Lemma 1.** *Assume $w^{(0)}$ is uniform on the unit sphere (equivalently, $w^{(0)} = g/\|g\|_2$ for $g \sim \mathcal{N}(0, I)$). For some $c_0 \in (0, \frac{\sqrt{3}}{2})$ and $p_{PZ}(c) := (1 - 1/\sqrt{2}) \cdot \frac{(1 - \frac{4}{3}c^2)^2}{3^8}$, for all $N \geq 1$ and all $d \geq 3$,*

$$\mathbf{Pr}\left[ |q^{(0)}| \geq \frac{c_0}{\sqrt{N}} \right] \geq p_{PZ}(c_0).$$

*As an example, we can choose $c_0 = \sqrt{3/8}$, in which case $p_{PZ}(c_0) = \frac{2 - \sqrt{2}}{8 \cdot 3^8}$.*

*Proof.* Fix $w = w^{(0)}$ and define a single-sample random variable $Z := y(x^\top w)^2$ so that $q^{(0)} = \frac{1}{N} \sum_{s=1}^{N} Z_s$ for i.i.d. copies $Z_s$.

**Step 1: conditional mean.**

$$\mu(w) := \mathbb{E}[Z \mid w] = \mathbb{E}\left[ x_1 x_2 \left( \sum_{i=1}^{d} w_i x_i \right)^2 \right] = \sum_{i,j} w_i w_j \mathbb{E}[x_1 x_2 x_i x_j] = 2w_1 w_2.$$

**Step 2: conditional variance is bounded below on a constant-probability event over $w$.** Since $y^2 \equiv 1$,

$$\mathbb{E}[Z^2 \mid w] = \mathbb{E}[(x^\top w)^4 \mid w] = 3\|w\|_2^4 - 2\sum_{i=1}^{d} w_i^4 \geq \|w\|_2^4 = 1.$$

Therefore,

$$\mathrm{Var}(Z \mid w) = \mathbb{E}[Z^2 \mid w] - \mu(w)^2 \geq 1 - 4w_1^2 w_2^2 \geq 1 - (w_1^2 + w_2^2)^2.$$

Define the event $\mathcal{E} := \{w_1^2 + w_2^2 \leq 1/2\}$. On $\mathcal{E}$ we have $\mathrm{Var}(Z \mid w) \geq 1 - 1/4 = 3/4$. Moreover, since $w$ is uniform on the sphere and $d \geq 3$, the random variable $w_1^2 + w_2^2$ has a $\mathrm{Beta}(1, (d-2)/2)$ distribution, hence

$$\mathbf{Pr}[\mathcal{E}] = \mathbf{Pr}[w_1^2 + w_2^2 \leq 1/2] = 1 - (1 - 1/2)^{(d-2)/2} \geq 1 - 2^{-1/2} =: p_1,$$

where $p_1 > 0$ is an absolute constant.

**Step 3: Paley-Zygmund on $(q^{(0)})^2$.** Condition on $w \in \mathcal{E}$. We already have $\sigma^2(w) := \mathrm{Var}(Z \mid w) \geq 3/4$, hence

$$\mathbb{E}[(q^{(0)})^2 \mid w] = \mu(w)^2 + \frac{\sigma^2(w)}{N} \geq \frac{3}{4N}.$$

We also need to upper bound the second moment of $(q^{(0)})^2$, i.e. a fourth-moment upper bound for $q^{(0)}$. For fixed $w$, the random variable $q^{(0)}$ is a polynomial of total degree at most 4 in the independent Rademacher variables $\{x_i^{(s)}\}_{i \in [d], s \in [N]}$ (after multilinearization using $x_i^2 \equiv 1$). By the Bonami-Beckner (hypercontractive) inequality, for any degree-$d$ polynomial $f$ of Rademachers,

$$(\mathbb{E}[|f|^4])^{1/4} \leq (4-1)^{d/2}(\mathbb{E}[|f|^2])^{1/2}$$

Applying this with $f = q^{(0)}$ (conditional on $w$) gives

$$\mathbb{E}[(q^{(0)})^4 \mid w] \leq 3^8 \mathbb{E}[(q^{(0)})^2 \mid w]^2. \tag{B.6}$$

Apply Paley-Zygmund to the non-negative random variable $Y := (q^{(0)})^2$ conditional on $w \in \mathcal{E}$:

$$\mathbf{Pr}\left[(q^{(0)})^2 \geq \theta \mathbb{E}[(q^{(0)})^2 \mid w] \,\Big|\, w\right] \geq \frac{(1-\theta)^2 \mathbb{E}[(q^{(0)})^2 \mid w]^2}{\mathbb{E}[(q^{(0)})^4 \mid w]} \geq \frac{(1-\theta)^2}{3^8},$$

where we used equation B.6. With $\theta = 1/2$,

$$\mathbf{Pr}\left[(q^{(0)})^2 \geq \theta \mathbb{E}[(q^{(0)})^2 \mid w] \,\Big|\, w\right] \geq \frac{1}{4 \cdot 3^8} =: p_2$$

On this event,

$$|q^{(0)}| \geq \sqrt{\tfrac{1}{2}\mathbb{E}[(q^{(0)})^2 \mid w]} \geq \sqrt{\frac{3}{8}} \cdot \frac{1}{\sqrt{N}}.$$

Thus, for $c_0 := \sqrt{3/8}$,

$$\mathbf{Pr}\left[|q^{(0)}| \geq \frac{c_0}{\sqrt{N}}\right] \geq \mathbf{Pr}[\mathcal{E}] \cdot p_2 \geq p_1 p_2 =: p_{ZL}(c_0).$$

$\square$

We also need the lucky event of $\mathrm{sign}(a^{(0)}) = \mathrm{sign}(q^{(0)})$ so that the updates don't flip the output weight's sign.

**Lemma 2.** *Assume $a^{(0)}$ is independent of $(w^{(0)}, \widehat{M})$, symmetric about 0, and that $\mathbf{Pr}[a^{(0)} = 0] = 0$. Then for every threshold $q_\star > 0$,*

$$\mathbf{Pr}\left[\mathrm{sign}(a^{(0)}) = \mathrm{sign}(q^{(0)}) \text{ and } |q^{(0)}| \geq q_\star\right] = \frac{1}{2}\mathbf{Pr}\left[|q^{(0)}| \geq q_\star\right].$$

*Proof.* Condition on $(w^{(0)}, \widehat{M})$ so that $q^{(0)}$ is fixed. On the event $q^{(0)} \neq 0$, symmetry and independence of $a^{(0)}$ imply $\mathbf{Pr}[\mathrm{sign}(a^{(0)}) = \mathrm{sign}(q^{(0)}) \mid q^{(0)}] = 1/2$. Multiply by $\mathbb{1}\{|q^{(0)}| \geq q_\star\}$ and average over $(w^{(0)}, \widehat{M})$. $\square$

B.2.2   STABILITY OF $q^{(t)}$ FOR SMALL $a^{(t)}$

Next, we show that under a stability condition of $\eta |a| \|\widehat{M}\|_2 \le 1/2$, the update in $q$ has the same sign as $a$.

**Lemma 3.** *Fix any unit vector $w \in \mathbb{R}^d$, scalar $a \in \mathbb{R}$, and learning rate $\eta > 0$. Define*

$$\tilde{w} := (I + \eta a \widehat{M})w, \qquad w^+ := \tilde{w}/\|\tilde{w}\|_2.$$

*Let $q := w^\top \widehat{M} w$ as before, and define $q^+ := (w^+)^\top \widehat{M} w^+$ similarly.*

*Then, under the stability condition of $\eta |a| \|\widehat{M}\|_2 \le 1/2$, $q^+ - q$ has the same sign as $a$ (or is zero).*

*Proof.* Define the following quantities

$$s := w^\top \widehat{M}^2 w, \qquad r := w^\top \widehat{M}^3 w. \tag{B.7}$$

With $\widehat{M}$ being symmetric, we have

$$q^+ = \frac{\tilde{w}^\top \widehat{M} \tilde{w}}{\tilde{w}^\top \tilde{w}} = \frac{w^\top (\widehat{M} + 2\eta a \widehat{M}^2 + \eta^2 a^2 \widehat{M}^3)w}{w^\top (I + 2\eta a \widehat{M} + \eta^2 a^2 \widehat{M}^2)w} = \frac{q + 2\eta a s + \eta^2 a^2 r}{1 + 2\eta a q + \eta^2 a^2 s},$$

and the update in $q$ is

$$q^+ - q = \frac{2\eta a(s - q^2) + \eta^2 a^2 (r - qs)}{1 + 2\eta a q + \eta^2 a^2 s}. \tag{B.8}$$

Note that the denominator in equation B.8 is positive, which follows from the stability assumption $\eta |a| \|\widehat{M}\|_2 \le 1/2$. The sign of $q^+ - q$ hence depends on the two terms in the numerator, which we bound separately below.

First, let $\widehat{M} = \sum_{i=1}^d \lambda_i u_i u_i^\top$ be an eigendecomposition and set $\alpha_i := \langle w, u_i \rangle^2$ so that $\alpha$ is a probability vector. Let $\Lambda$ be the random variable taking value $\lambda_i$ with probability $\alpha_i$. Then

$$q = \sum_i \alpha_i \lambda_i = \mathbb{E}[\Lambda], \qquad s = \sum_i \alpha_i \lambda_i^2 = \mathbb{E}[\Lambda^2], \qquad r = \sum_i \alpha_i \lambda_i^3.$$

This directly gives that $s - q^2 = \text{Var}(\Lambda) \ge 0$.

For the second term, note that

$$r - qs = \mathbb{E}[\Lambda^3] - \mathbb{E}[\Lambda]\mathbb{E}[\Lambda^2] = \mathbb{E}\big[(\Lambda - \mathbb{E}[\Lambda])^2(\Lambda + \mathbb{E}[\Lambda])\big].$$

Since $|\Lambda + \mathbb{E}[\Lambda]| \le 2\|\widehat{M}\|_2$, we obtain

$$|r - qs| \le 2\|\widehat{M}\|_2 \mathbb{E}[(\Lambda - \mathbb{E}[\Lambda])^2] = 2\|\widehat{M}\|_2(s - q^2),$$

Combining this with the stability assumption of $\eta |a| \|\widehat{M}\|_2 \le 1/2$, we can bound the second term of the numerator in Equation (B.8) by

$$\big|\eta^2 a^2 (r - qs)\big| \le 2\eta^2 |a|^2 \|\widehat{M}\|_2 (s - q^2) \le \eta |a|(s - q^2).$$

Therefore,

- if $a \ge 0$, then $2\eta a(s - q^2) + \eta^2 a^2(r - qs) \ge 2\eta a(s - q^2) - \eta a(s - q^2) = \eta a(s - q^2) \ge 0$;

- if $a \le 0$, then $2\eta a(s - q^2) + \eta^2 a^2(r - qs) \le 2\eta a(s - q^2) + \eta |a|(s - q^2) = \eta a(s - q^2) \le 0$.

Thus the numerator in equation B.8 and hence $q^+ - q$ has the same sign as $a$ (or is zero).   □

### B.2.3 Linear growth of $a^{(t)}$

Next, we show that $a$ grows linearly when conditioned on the lucky event in Lemma 2 and the stability assumption in Lemma 3, from which an upper bound on $T_\star$ (i.e., time for $a$ to grow to $a_\star$) directly follows.

**Lemma 4.** *Assume the initialization event*

$$\text{sign}(a^{(0)}) = \text{sign}(q^{(0)}) \qquad \text{and} \qquad |q^{(0)}| \geq q_\star > 0,$$

*for $q_\star := \frac{c_0}{\sqrt{N}}$ from Lemma 1. Further, assume the stability condition*

$$\eta a_\star \|\widehat{M}\|_2 \leq \frac{1}{2}.$$

*Then for all $t < T_\star$ we have $\text{sign}(a^{(t)}) = \text{sign}(q^{(t)}) = \text{sign}(q^{(0)})$ and $|q^{(t)}| \geq |q^{(0)}| \geq q_\star$. Consequently,*

$$T_\star \leq \left\lceil \frac{2(a_\star - |a^{(0)}|)_+}{\eta q_\star} \right\rceil. \tag{B.9}$$

*Proof.* Fix any $t < T_\star$. Since $|a^{(t)}| \leq a_\star$ and $\eta a_\star \|\widehat{M}\|_2 \leq 1/2$, we may apply Lemma 3 to the inner weight update at time $t$. It implies that $q^{(t+1)} - q^{(t)}$ has the same sign as $a^{(t)}$ (or is zero).

We next show by induction that $\text{sign}(a^{(t)}) = \text{sign}(q^{(t)}) = \text{sign}(q^{(0)})$ and $|q^{(t)}| \geq |q^{(0)}|$ for all $t < T_\star$. The base case $t = 0$ holds by assumption. Assume it holds at time $t$. Because $t < T_\star$ we have $|a^{(t)}| < 1$, so clipping is inactive and

$$a^{(t+1)} = a^{(t)} + \frac{\eta}{2}q^{(t)}.$$

Since $\text{sign}(a^{(t)}) = \text{sign}(q^{(t)})$, we get $\text{sign}(a^{(t+1)}) = \text{sign}(a^{(t)})$ and

$$|a^{(t+1)}| = |a^{(t)}| + \frac{\eta}{2}|q^{(t)}|.$$

Thus $\text{sign}(a^{(t)})$ remains constant and equal to $\text{sign}(q^{(0)})$ throughout $t < T_\star$. Returning to Lemma 3, this means $q^{(t)}$ is pushed monotonically in the direction of $\text{sign}(a^{(t)}) = \text{sign}(q^{(0)})$ and therefore cannot cross 0. Hence $\text{sign}(q^{(t)}) = \text{sign}(q^{(0)})$ and $|q^{(t)}| \geq |q^{(0)}| \geq q_\star$ for all $t < T_\star$.

Finally, using $|q^{(t)}| \geq q_\star$ gives the linear growth bound

$$|a^{(t+1)}| = |a^{(t)}| + \frac{\eta}{2}|q^{(t)}| \geq |a^{(t)}| + \frac{\eta}{2}q_\star,$$

so $|a^{(t)}| \geq |a^{(0)}| + t \cdot \frac{\eta}{2}q_\star$ while $t < T_\star$. Solving for the first $t$ such that $|a^{(t)}| \geq a_\star$ yields equation B.9. $\qquad\square$

### B.2.4 Choosing largest stable $a_\star$

In order to find a bound on how large we can set $a_\star$, we will first bound $\|\widehat{M}\|_2$.

**Lemma 5** (Matrix Bernstein bound for $\|\widehat{M}\|_2$). *For every $\delta \in (0,1)$, with probability at least $1 - \delta$,*

$$\|\widehat{M}\|_2 \leq 1 + C\left(\sqrt{\frac{d\log(2d/\delta)}{N}} + \frac{d\log(2d/\delta)}{N}\right)$$

*for a universal constant $C > 0$.*

*Proof.* Let $A_s := y^{(s)}x^{(s)}x^{(s)\top}$. Since $A_s^2 = (x^{(s)}x^{(s)\top})^2 = \|x^{(s)}\|_2^2 x^{(s)}x^{(s)\top} = dx^{(s)}x^{(s)\top}$ (and $y^{(s)2} = 1$), we have

$$\mathbb{E}[A_s^2] = d\mathbb{E}[xx^\top] = dI.$$

Write the population matrix as

$$M := \mathbb{E}[A_s] = \mathbb{E}[yxx^\top] = e_1 e_2^\top + e_2 e_1^\top, \qquad \|M\|_2 = 1.$$

Define centered summands $X_s := A_s - M$ so that $\mathbb{E}[X_s] = 0$ and

$$\widehat{M} - M = \frac{1}{N}\sum_{s=1}^{N} X_s.$$

We bound $\|X_s\|_2 \le \|A_s\|_2 + \|M\|_2 \le d + 1 \le 2d$, so we may take $R := 2d$.

$$\mathbb{E}[X_s^2] = \mathbb{E}[(A_s - M)^2] = \mathbb{E}[A_s^2] - M^2 \preceq \mathbb{E}[A_s^2] = dI,$$

hence

$$\sigma^2 := \left\|\sum_{s=1}^{N} \mathbb{E}[X_s^2]\right\|_2 \le Nd.$$

Matrix Bernstein (for sums of independent mean-zero self-adjoint matrices) then yields that with probability at least $1 - \delta$,

$$\left\|\sum_{s=1}^{N} X_s\right\|_2 \le C\left(\sqrt{\sigma^2 \log(2d/\delta)} + R\log(2d/\delta)\right) \le C\left(\sqrt{Nd\log(2d/\delta)} + d\log(2d/\delta)\right).$$

Dividing by $N$ gives

$$\|\widehat{M} - M\|_2 \le C\left(\sqrt{\frac{d\log(2d/\delta)}{N}} + \frac{d\log(2d/\delta)}{N}\right).$$

Finally, $\|\widehat{M}\|_2 \le \|M\|_2 + \|\widehat{M} - M\|_2$ and $\|M\|_2 = 1$, proving the claim. $\qquad\square$

Substituting the above gives the following.

**Corollary 3.** *Fix* $\delta \in (0,1)$ *and stepsize* $\eta > 0$. *Let* $B_{N,d,\delta} = 1 + C\left(\sqrt{\frac{d\log(2d/\delta)}{N}} + \frac{d\log(2d/\delta)}{N}\right)$. *If*

$$a_\star \le \min\left\{1, \frac{1}{2\eta\, B_{N,d,\delta}}\right\},$$

*then with probability at least* $1 - \delta$ *the stability event*

$$\eta\,|a^{(t)}|\,\|\widehat{M}\|_2 \le \tfrac{1}{2} \qquad \text{for all } t < T_\star := \min\{t : |a^{(t)}| \ge a_\star\}$$

*holds.*

**Corollary 4** (Bound for random label $\|\widehat{M}\|_2$)**.** *For $y$ uniformly sampled from $\{-1, +1\}$, for any $\delta \in (0,1)$, with probability at least $1 - \delta$,*

$$\|\widehat{M}\|_2 \le O\left(\sqrt{\frac{d\log(2d/\delta)}{N}} + \frac{d\log(2d/\delta)}{N}\right)$$

*Proof.* The calculation is the same as Lemma 5, except for $M = 0$ because $y$ is uniformly random. $\qquad\square$

## B.3 ANALYSIS OF PHASE 2

In Phase 2, we replace $\widehat{M}$ by the population matrix $M = e_1 e_2^\top + e_2 e_1^\top$. Its spectrum is explicit: let

$$u_+ := \frac{e_1 + e_2}{\sqrt{2}}, \qquad u_- := \frac{e_1 - e_2}{\sqrt{2}},$$

then $Mu_+ = u_+$, $Mu_- = -u_-$, and $Mv = 0$ for all $v \perp \text{span}\{e_1, e_2\}$.

We show that in Phase 2, $w$ converges quickly to one of $u_+, u_-$ following a power iteration on $M$. [7]

**Lemma 6** (Population contraction). *Assume* $\eta \leq \frac{1}{2}$, $a^{(0)} \neq 0$, *and* $\text{sign}(a^{(0)}) = \text{sign}(q^{(0)})$. *Consider projected updates*

$$a^{(t+1)} = \text{clip}_{[-1,1]}\left(a^{(t)} + \tfrac{\eta}{2} q^{(t)}\right), \qquad w^{(t+1)} = \frac{w^{(t)} + \eta a^{(t)} M w^{(t)}}{\|w^{(t)} + \eta a^{(t)} M w^{(t)}\|_2}, \qquad q^{(t)} := (w^{(t)})^\top M w^{(t)}.$$

*Let*

$$u_a := \frac{e_1 + \text{sign}(a^{(0)}) e_2}{\sqrt{2}}, \quad u_{-a} := \frac{e_1 - \text{sign}(a^{(0)}) e_2}{\sqrt{2}}, \quad \alpha_t := |\langle w^{(t)}, u_a \rangle|, \quad r_t := \frac{\sqrt{1 - \alpha_t^2}}{\alpha_t}.$$

*Then:*

1. **Sign stability and monotonicity.** *For all* $t \geq 0$, $\text{sign}(a^{(t)}) = \text{sign}(q^{(t)}) = \text{sign}(a^{(0)})$, *and* $|a^{(t)}|$ *is non-decreasing.*

2. **Alignment contraction.** *For all* $t \geq 0$,

$$r_{t+1} \leq \frac{1}{1 + \eta|a^{(t)}|} r_t \leq \frac{1}{1 + \eta|a^{(0)}|} r_t.$$

*Consequently, after*

$$T_2 := \left\lceil \frac{2}{\eta|a^{(0)}|} \log\left(\frac{1}{\alpha_0^2 \varepsilon}\right) \right\rceil$$

*steps we have* $\alpha_{T_2}^2 \geq 1 - \varepsilon$ *for any* $\varepsilon \in (0, 1/2)$.

*Proof.* Because $\eta \leq 1/2$ and $|a^{(t)}| \leq 1$, the stability condition of Lemma 3 (i.e., $\eta|a^{(t)}|\|M\|_2 \leq \frac{1}{2}$) holds with $\widehat{M} = M$ for every step. Hence $q^{(t+1)} - q^{(t)}$ has the same sign as $a^{(t)}$ (or is 0). Since $\text{sign}(a^{(0)}) = \text{sign}(q^{(0)})$ and

$$a^{(t+1)} = a^{(t)} + \tfrac{\eta}{2} q^{(t)} \quad \text{as long as clipping is inactive,}$$

the signs of $a^{(t)}$ and $q^{(t)}$ cannot flip; moreover $|a^{(t)}|$ is nondecreasing, and clipping preserves the sign once $|a^{(t)}|$ hits 1. This proves (1).

For (2), decompose $w^{(t)}$ as

$$w^{(t)} = c_t u_a + b_t u_{-a} + v_t,$$

where $v_t \perp \text{span}\{e_1, e_2\}$. Then $(I + \eta a M)u_a = (1 + \eta|a|)u_a$, $(I + \eta a M)u_{-a} = (1 - \eta|a|)u_{-a}$, and $(I + \eta a M)v = v$. Thus before normalization,

$$(I + \eta a M)w^{(t)} = (1 + \eta|a|)c_t u_a + (1 - \eta|a|)b_t u_{-a} + v_t.$$

---

[7]The upper bound on $T_2$ is likely improvable to $O(\log(1/a^{(0)}))$.

After normalization, ratios between the orthogonal component and the $u_a$ component is

$$r_{t+1} = \frac{\sqrt{b_{t+1}^2 + \|v_{t+1}\|^2}}{|c_{t+1}|} = \frac{\sqrt{(1 - \eta|a^{(t)}|)^2 b_t^2 + \|v_t\|^2}}{(1 + \eta|a^{(t)}|)|c_t|} \le \frac{\sqrt{b_t^2 + \|v_t\|^2}}{(1 + \eta|a^{(t)}|)|c_t|} = \frac{r_t}{1 + \eta|a^{(t)}|}.$$
(B.10)

Combined with (1), this shows that $r_t$ contracts by at least a factor of $\frac{1}{1+\eta|a^{(0)}|}$, which is strictly smaller than 1 since $a^{(0)} \neq 0$.

The convergence time $T_2$ follows from $\log(1 + \eta|a|) \ge \frac{\eta|a|}{1+\eta|a|} \ge \frac{\eta|a|}{2}$, since $\eta|a| \le 1$.

$\square$

## B.4   COMBINING BOTH PHASES

We have shown that $T_\star \le \left\lceil \frac{2(a_\star - |a^{(0)}|)}{\eta q_\star} \right\rceil$ (Lemma 4) and $T_2 \le \frac{2}{\eta|a_\star|} \log\left(\frac{1}{\alpha_0^2 \varepsilon}\right)$ (Lemma 6). To reason about the overall time $T_\star + T_2$, it remains to check how $\alpha_0$ depends on $a_\star$, which in turn depends on how much $w$ moves during Phase 1.

In the following, we will first bound $w$'s drift (Lemma 7) which will then allow us to relate $a_\star$ and $\alpha_0$ (Lemma 8), and present the final convergence bound in Appendix B.4.3.

### B.4.1   $w^{(t)}$ GROWS SLOWLY

We first need to bound how much $w$ drifts in phase 1. We show that under the assumptions of Lemma 4, the input weight $w^{(t)}$ changes little up to time $T_\star$.

**Lemma 7** (Control of inner weight drift up to $T_\star$). *Under the assumptions of Lemma 4,*

$$\|w^{(T_\star)} - w^{(0)}\|_2 \le \frac{8\|\widehat{M}\|_2}{q_\star} a_\star(a_\star - |a^{(0)}|)_+ + 4\eta\|\widehat{M}\|_2 a_\star.$$
(B.11)

*Proof.* Write the pre-normalization iterate as

$$\tilde{w}^{(t+1)} = w^{(t)} + \eta a^{(t)} \widehat{M} w^{(t)}, \qquad w^{(t+1)} = \tilde{w}^{(t+1)}/\|\tilde{w}^{(t+1)}\|_2.$$

Let $u^{(t)} := \eta a^{(t)} \widehat{M} w^{(t)}$, so $\tilde{w}^{(t+1)} = w^{(t)} + u^{(t)}$. For $t < T_\star$ we have $|a^{(t)}| \le a_\star$, hence

$$\|u^{(t)}\|_2 \le \eta|a^{(t)}| \|\widehat{M}\|_2 \le \eta a_\star \|\widehat{M}\|_2 \le \frac{1}{2}.$$

For any unit vector $w$ and any $u$ with $\|u\|_2 \le 1/2$, one has the standard normalization Lipschitz bound

$$\left\| \frac{w + u}{\|w + u\|_2} - w \right\|_2 \le 4\|u\|_2,$$

which we apply with $(w, u) = (w^{(t)}, u^{(t)})$ to get

$$\|w^{(t+1)} - w^{(t)}\|_2 \le 4\|u^{(t)}\|_2 \le 4\eta|a^{(t)}| \|\widehat{M}\|_2.$$

Summing over $t = 0, 1, \ldots, T_\star - 1$ yields

$$\|w^{(T_\star)} - w^{(0)}\|_2 \le 4\eta\|\widehat{M}\|_2 \sum_{t < T_\star} |a^{(t)}|.$$

Using the crude bound $\sum_{t < T_\star} |a^{(t)}| \le T_\star a_\star$, which is sufficient for the final scaling, together with equation B.9 gives

$$\sum_{t < T_\star} |a^{(t)}| \le a_\star \left( \frac{2(a_\star - |a^{(0)}|)_+}{\eta q_\star} + 1 \right) = \frac{2a_\star(a_\star - |a^{(0)}|)_+}{\eta q_\star} + a_\star.$$

Plugging in yields

$$\|w^{(T_\star)} - w^{(0)}\|_2 \leq 4\eta\|\widehat{M}\|_2 \left( \frac{2a_\star(a_\star - |a^{(0)}|)_+}{\eta q_\star} + a_\star \right) = \frac{8\|\widehat{M}\|_2}{q_\star} a_\star(a_\star - |a^{(0)}|)_+ + 4\eta\|\widehat{M}\|_2 a_\star,$$

which is equation B.11. $\qquad\square$

### B.4.2 Connecting $\alpha_0$ and $a_\star$

**Lemma 8** (Lower bound on $\alpha_0^2$ in terms of $a_\star$). *Let* $u := (e_1 + \text{sign}(a^{(T_\star)})e_2)/\sqrt{2}$ *and define*

$$\alpha_0 := |\langle w^{(T_\star)}, u \rangle|.$$

*On the event* $\|w^{(T_\star)} - w^{(0)}\|_2 \leq \varepsilon_{\text{drift}}$, *we have*

$$\alpha_0^2 \geq (|\langle w^{(0)}, u \rangle| - \varepsilon_{\text{drift}})_+^2.$$

*Under the assumptions of Lemma 7, we may take*

$$\varepsilon_{\text{drift}} := \frac{8\|\widehat{M}\|_2}{q_\star} a_\star(a_\star - |a^{(0)}|)_+ + 4\eta\|\widehat{M}\|_2 a_\star,$$

*so* $\alpha_0^2$ *is explicitly lower bounded in terms of* $a_\star$.

*Proof.* By Cauchy–Schwarz, $|\langle w^{(T_\star)}, u \rangle - \langle w^{(0)}, u \rangle| \leq \|w^{(T_\star)} - w^{(0)}\|_2\|u\|_2 = \|w^{(T_\star)} - w^{(0)}\|_2$. This implies $|\langle w^{(T_\star)}, u \rangle| \geq |\langle w^{(0)}, u \rangle| - \varepsilon_{\text{drift}}$ on the event. The explicit choice of $\varepsilon_{\text{drift}}$ is equation B.11. $\qquad\square$

**Lemma 9** (Constant-probability lower bound on random initialization alignment). *Let* $u \in \mathbb{R}^d$ *be any fixed unit vector and let* $w^{(0)}$ *be uniform on the unit sphere. Then there exists a universal constant* $p_{\text{align}} > 0$ *such that for all* $d \geq 2$,

$$\mathbf{Pr}\left[ |\langle w^{(0)}, u \rangle| \geq \frac{1}{2\sqrt{d}} \right] \geq p_{\text{align}}.$$

*Proof.* Write $w^{(0)} = g/\|g\|_2$ for $g \sim \mathcal{N}(0, I)$ and rotate so that $u = e_1$. Then $|\langle w^{(0)}, u \rangle| = |g_1|/\|g\|_2$. On the event $\{|g_1| \geq 1\} \cap \{\|g\|_2 \leq 2\sqrt{d}\}$ we have $|g_1|/\|g\|_2 \geq 1/(2\sqrt{d})$. $\mathbf{Pr}\left[ \{|g_1| \geq 1\} \cap \{\|g\|_2 \leq 2\sqrt{d}\} \right] \geq \mathbf{Pr}\left[|g_1| \geq 1\right] - \mathbf{Pr}\left[\|g\|_2 > 2\sqrt{d}\right]$. The former one has constant probability and the latter one decays exponentially with $d$, so the intersection has probability at least some universal constant $p_{\text{align}} > 0$. $\qquad\square$

**Lemma 10** (Constant-probability simultaneous phase-1 bootstrap and population sign alignment). *Let*

$$u_\pm := \frac{e_1 \pm e_2}{\sqrt{2}}, \qquad P_{12} := u_+u_+^\top + u_-u_-^\top, \qquad q_{\text{pop}}(w) := w^\top M w.$$

*There exist universal constants* $c_0 > 0$ *and* $p_{\text{all}} > 0$ *such that, with*

$$\mathcal{P}_0 := \left\{ |q_{\text{pop}}(w^{(0)})| \geq \frac{3}{4d}, \quad \|P_{12}w^{(0)}\|_2 \leq \frac{1}{\sqrt{d}} \right\}$$

*and*

$$\mathcal{G}_{\text{sign}} := \left\{ \text{sign}(a^{(0)}) = \text{sign}(q^{(0)}) = \text{sign}(q_{\text{pop}}(w^{(0)})), \quad |q^{(0)}| \geq \frac{c_0}{\sqrt{N}} \right\},$$

*we have*

$$\mathbf{Pr}\left[ \mathcal{G}_{\text{sign}} \cap \mathcal{P}_0 \right] \geq p_{\text{all}}.$$

*In particular, on this event, the empirical sign used to grow $a$ in phase 1 is already the population sign that will be needed at the start of phase 2.*

*Proof.* Write $z_\pm := \langle w^{(0)}, u_\pm \rangle$ and $r^2 := z_+^2 + z_-^2 = \|P_{12}w^{(0)}\|_2^2$. Conditional on $r$, the angle $(z_+, z_-)/r$ is uniform on the unit circle, and $r^2 \sim \text{Beta}(1, (d-2)/2)$. Consider the event

$$\mathcal{R} := \left\{ \frac{9}{10d} \leq r^2 \leq \frac{1}{d}, \quad |\cos(2\theta)| \geq \frac{5}{6} \right\}, \qquad (z_+, z_-) = r(\cos\theta, \sin\theta).$$

On $\mathcal{R}$,

$$|q_{\text{pop}}(w^{(0)})| = |z_+^2 - z_-^2| = r^2 |\cos(2\theta)| \geq \frac{3}{4d}, \qquad \|P_{12}w^{(0)}\|_2 = r \leq \frac{1}{\sqrt{d}},$$

so $\mathcal{R} \subseteq \mathcal{P}_0$. The radial probability of $\{9/(10d) \leq r^2 \leq 1/d\}$ is bounded below by a universal constant for all $d \geq 3$ after decreasing the constant to cover the finitely many small dimensions, and the angular event $\{|\cos(2\theta)| \geq 5/6\}$ also has universal positive probability. Hence $\mathbf{Pr}[\mathcal{P}_0] \geq p_{\text{pop}} > 0$.

Fix any $w \in \mathcal{P}_0$ and set $\mu := q_{\text{pop}}(w)$. For one phase–1 sample, let $Z := y(x^\top w)^2$, so that $\mathbb{E}[Z \mid w] = \mu$ and $q^{(0)} = N^{-1}\sum_{s=1}^N Z_s$. Since $\|P_{12}w\|_2^2 \leq 1/d \leq 1/2$, the variance lower bound in Lemma 1 gives $\text{Var}(Z \mid w) \geq 3/4$. The same hypercontractive fourth-moment bound used in Lemma 1, applied to $\sqrt{N}(q^{(0)} - \mu)$, gives a universal fourth-moment upper bound.

We use the following elementary one-sided consequence of these two moment bounds: if $X$ is mean zero, $\mathbb{E}[X^2] = \sigma^2$, and $\mathbb{E}[X^4] \leq K\sigma^4$, then there are constants $c_K, p_K > 0$ depending only on $K$ such that $\mathbf{Pr}[X \geq c_K\sigma] \geq p_K$. Indeed, writing $X_+ = \max\{X, 0\}$ and $X_- = \max\{-X, 0\}$, the identity $\mathbb{E}[X_+] = \mathbb{E}[X_-]$ and interpolation between $L_1, L_2, L_4$ norms give $\mathbb{E}[X_+] \geq \sigma/(2^{3/2}\sqrt{K})$. Therefore, with $\theta := 1/(2^{5/2}\sqrt{K})$,

$$\mathbb{E}[X_+] \leq \theta\sigma + \left(\mathbb{E}[X_+^2]\right)^{1/2}\mathbf{Pr}[X \geq \theta\sigma]^{1/2} \leq \theta\sigma + \sigma\,\mathbf{Pr}[X \geq \theta\sigma]^{1/2},$$

which implies $\mathbf{Pr}[X \geq \theta\sigma] \geq \theta^2$. Thus one may take $c_K = \theta$ and $p_K = \theta^2$. Applying this to $X := \text{sign}(\mu)\sqrt{N}(q^{(0)} - \mu)$, and using $\text{sign}(\mu)\sqrt{N}\,q^{(0)} = X + |\mu|\sqrt{N} \geq X$, gives, after decreasing $c_0$ if necessary, a universal $p_{\text{one}} > 0$ such that

$$\mathbf{Pr}\left[\text{sign}(\mu)\,q^{(0)} \geq \frac{c_0}{\sqrt{N}} \;\middle|\; w^{(0)} = w\right] \geq p_{\text{one}}.$$

Equivalently, conditional on $w \in \mathcal{P}_0$, the empirical quadratic form has the same sign as the population quadratic form and has magnitude at least $c_0/\sqrt{N}$ with probability at least $p_{\text{one}}$. Finally, $a^{(0)}$ is independent of the samples and is symmetric about zero, so conditional on $(w^{(0)}, \widehat{M})$ the event $\text{sign}(a^{(0)}) = \text{sign}(q^{(0)})$ contributes an additional factor $1/2$. Therefore

$$\mathbf{Pr}\left[\mathcal{G}_{\text{sign}} \cap \mathcal{P}_0\right] \geq \frac{1}{2}p_{\text{pop}}p_{\text{one}} := p_{\text{all}} > 0.$$

$\square$

**Lemma 11** (Phase-1 drift preserves the population sign). *Define $q_{\text{pop}}(w) := w^\top M w$ and $P_{12} := u_+u_+^\top + u_-u_-^\top$ as in Lemma 10. Suppose*

$$|q_{\text{pop}}(w^{(0)})| \geq \frac{3}{4d}, \qquad \|P_{12}w^{(0)}\|_2 \leq \frac{1}{\sqrt{d}}, \qquad \|w^{(T_\star)} - w^{(0)}\|_2 \leq \frac{1}{4\sqrt{d}}.$$

*Then*

$$\text{sign}\left(q_{\text{pop}}(w^{(T_\star)})\right) = \text{sign}\left(q_{\text{pop}}(w^{(0)})\right).$$

*Moreover, if $s := \text{sign}(q_{\text{pop}}(w^{(0)}))$ and $u_s := (e_1 + se_2)/\sqrt{2}$, then*

$$|\langle w^{(T_\star)}, u_s \rangle| \geq \left(\frac{\sqrt{3}}{2} - \frac{1}{4}\right)\frac{1}{\sqrt{d}} \geq \frac{1}{2\sqrt{d}}.$$

*Proof.* Let $\Delta := w^{(T_\star)} - w^{(0)}$. Since $\|M\|_2 = 1$ and $\|Mw^{(0)}\|_2 = \|P_{12}w^{(0)}\|_2$, we have

$$\left| q_{\text{pop}}(w^{(T_\star)}) - q_{\text{pop}}(w^{(0)}) \right| = \left| 2(w^{(0)})^\top M \Delta + \Delta^\top M \Delta \right|$$
$$\leq 2\|P_{12}w^{(0)}\|_2 \|\Delta\|_2 + \|\Delta\|_2^2$$
$$\leq \frac{1}{2d} + \frac{1}{16d} = \frac{9}{16d} < \frac{3}{4d}.$$

Thus the perturbation is smaller than the initial population margin, so the sign of $q_{\text{pop}}$ is preserved.

For the alignment claim, write $z_s := \langle w^{(0)}, u_s \rangle$ and $z_{-s} := \langle w^{(0)}, u_{-s} \rangle$. Since $s\, q_{\text{pop}}(w^{(0)}) = z_s^2 - z_{-s}^2 \geq 3/(4d)$, we have $|z_s| \geq \sqrt{3}/(2\sqrt{d})$. Cauchy-Schwarz gives

$$|\langle w^{(T_\star)}, u_s \rangle| \geq |\langle w^{(0)}, u_s \rangle| - \|w^{(T_\star)} - w^{(0)}\|_2 \geq \left( \frac{\sqrt{3}}{2} - \frac{1}{4} \right) \frac{1}{\sqrt{d}} \geq \frac{1}{2\sqrt{d}}.$$

$\square$

### B.4.3 FINAL CONVERGENCE BOUND

Fix $\varepsilon \in (0, 1/2)$ and $\delta \in (0, 1)$. Run phase 1 on a fixed batch of size $N$ using updates equation B.2–equation B.3 until time

$$T_1 := \min\{t : |a^{(t)}| \geq a_\star\}.$$

Let $p_{\text{all}} > 0$ be the universal constant from Lemma 10. Then, by intersecting Lemma 10 with the operator-norm event $\{\|\widehat{M}\|_2 \leq B_{N,d,\delta}\}$, we get

$$\mathbf{Pr}\left[ \mathcal{G}_{\text{sign}} \cap \mathcal{P}_0 \cap \{\|\widehat{M}\|_2 \leq B_{N,d,\delta}\} \right] \geq p_{\text{all}} - \delta.$$

On this event, the following hold:

1. **Phase 1 time.** We have

$$T_1 \leq \left\lceil \frac{2(a_\star - |a^{(0)}|)_+}{\eta\, q_\star} \right\rceil \leq \left\lceil \frac{2a_\star}{\eta\, c_0} \sqrt{N} \right\rceil.$$

2. **Phase 2 alignment.** Switch to population gradients ($\widehat{M} \leftarrow M$), and run the population contraction on $w$ for

$$T_2 := \left\lceil \frac{2}{\eta a_\star} \log\left( \frac{16d}{\varepsilon} \right) \right\rceil$$

steps.

By Lemma 7 and the above choice of $a_\star$, we have

$$\|w^{(T_1)} - w^{(0)}\|_2 \leq \varepsilon_{\text{drift}} \leq \frac{1}{4\sqrt{d}}.$$

Since $\mathcal{G}_{\text{sign}}$ gives $\text{sign}(a^{(0)}) = \text{sign}(q_{\text{pop}}(w^{(0)}))$ and phase 1 preserves $\text{sign}(a^{(t)})$ up to $T_1$, Lemma 11 gives the missing population sign condition

$$\text{sign}(a^{(T_1)}) = \text{sign}\left( (w^{(T_1)})^\top M w^{(T_1)} \right).$$

The same lemma also gives

$$\alpha_0^2 := \left| \left\langle w^{(T_1)}, \frac{e_1 + \text{sign}(a^{(T_1)})e_2}{\sqrt{2}} \right\rangle \right|^2 \geq \frac{1}{4d}.$$

Therefore Lemma 6, applied from the phase–2 starting point and using $|a^{(T_1)}| \geq a_\star$, yields $\alpha_{T_2}^2 \geq 1 - \varepsilon$, where $\alpha_t := |\langle w^{(t)}, u \rangle|$ with $u = (e_1 + \text{sign}(a^{(T_1)})e_2)/\sqrt{2}$.

**Interpreting the Result** Let

$$B_{N,d,\delta} := 1 + C \left( \sqrt{\frac{d \log(2d/\delta)}{N}} + \frac{d \log(2d/\delta)}{N} \right)$$

be the deterministic bound from Lemma 5, so that $\mathbf{Pr}[\|\widehat{M}\|_2 \leq B_{N,d,\delta}] \geq 1 - \delta$. Let $c_0 > 0$ be the universal constant from Lemma 10, chosen small enough that Lemma 1 also applies, and set $q_\star := c_0/\sqrt{N}$. Choose

$$a_\star := \min \left\{ 1, \ \frac{1}{2\eta B_{N,d,\delta}}, \ \frac{1}{32\eta B_{N,d,\delta}\sqrt{d}}, \ \sqrt{\frac{c_0}{64\, B_{N,d,\delta}\sqrt{N}\sqrt{d}}} \right\}.$$

In particular, for $d \leq N \leq d^2$,

$$a_\star = \sqrt{\frac{c_0}{64\, B_{N,d,\delta}\sqrt{N}\sqrt{d}}} = O\left(\frac{1}{(Nd)^{1/4}}\right),$$

which yields

$$T_1 \lesssim \frac{N^{1/4}}{\eta d^{1/4}}, \qquad T_2 \lesssim \frac{(Nd)^{1/4}}{\eta} \log\left(\frac{d}{\varepsilon}\right). \tag{B.12}$$

The total number of steps needed decreases as $N$ gets smaller. The gain of the two-phase schedule is that it avoids entering phase 2 with a *too small* outer gain $|a|$: since the $w$-update is scaled by $a_t$, small $|a_t|$ slows representation learning even if $w_0$ has typical random alignment. Phase 1 increases $|a_t|$ using the stronger fixed-batch bootstrap signal $|q^{(0)}| \sim 1/\sqrt{N}$, whereas the analogous population bootstrap signal at random initialization is only $|w^\top M w| = \Theta(1/d)$.

## C  EXPERIMENT DETAILS

**Tasks** We consider synthetic tasks, which have tunable parameters and thus allow for explicit control over task complexity.

We start with two classic feature learning which have been extensively studied in the literature.

- **Single-index model (SIM)**: the input is a Gaussian vector $x \sim \mathcal{N}(0, I_d)$, and the label is given by $y := \phi(\langle w^*, x \rangle)$, where $w^*$ is the ground truth feature vector, and $\phi : \mathbb{R} \to \mathbb{R}$ is an unknown link function. Our experiments take the link function to be a Hermite polynomial, denoted as $\mathrm{He}_k$ for some order $k$.

- $(d, k)$-**sparse parity**: the input is a boolean vector $x \sim \mathrm{Unif}(\{\pm 1\}^d)$, and the label is given by $y := \prod_{i \in S} x_i$, where $S \subset [d]$ is an unknown support of size $k$.

We consider two additional algorithmic tasks for Transformers:

- **In-context linear regression**: the input is a sequence $x_1, y_1, x_2, y_2, \ldots, x_k, y_k, x_q$ of length $2k + 1$, where each sequence we independently sample a $w \sim \mathcal{N}(0, I_n)$, $x_i \sim \mathcal{N}(0, I_n)$, $y_i = w^\top x_i, \forall i \in [k]$ and the label is given by $y := w^\top x_q$.

- $(N, p)$-**modular addition**: the input are two numbers $x, z \sim \mathrm{Unif}([N])$, and the label is given by $y := (x + z) \mod p$ for some prime $p$. For Transformer experiments, $x, z$ are each represented by $\lceil \log_b N \rceil$ digits in base-$b$, and the output logits have size $p$.

**Experimental setup** Our primary focus is performance under a given compute, which is measured by the batch size $\times$ number of optimization steps. We report *expected performance* under a fixed compute by taking the accuracy or loss averaged over random seeds. For

tasks where the accuracy exhibits shape phase transitions, the average accuracy can also be interpreted as the *probability of success*.

We train with both MLPs and Transformers. The MLP has ReLU activation, no output bias, and no residual connections. The Transformer has an optional QK normalization. Models are of depth-2 unless specified otherwise. All weights are initialized with Pytorch defaults; for example, $W_{ij} \sim \text{Unif}[-1/\sqrt{d_{\text{in}}}, 1/\sqrt{d_{\text{in}}}]$. [8] We use the SGD optimizer for MLPs and AdamW for Transformers, and sweep over the learning rate for each setup.

## C.1 DATA REUSE STRATEGIES

We consider both batch stochastic gradient descent (SGD) and (full-batch) gradient descent (GD) over datasets of different sizes. For batch SGD, the batches are sampled uniformly over the distribution with replacement. Note that when the number of steps is more than the number of samples, we can choose to sample with or without replacement. Despite the apparent similarity, these two strategies correspond to different algorithms. For example, for linear regression, the former is equivalent (in terms of sample complexity) to gradient descent (Lin et al., 2025), whereas the latter is closer to online SGD which can be better or worse than GD depending on the problem structure (Wu et al., 2025). In our experiments though, we do not notice an empirical difference between the two based, hence we use them interchangeably.

We additionally consider **multi-phase training**, where the dataset sizes across phases can vary. In particular, for $T$-phase repeat, batches are sampled from a subset $\mathcal{S}_i$ at the $i_{th}$ stage for $i \in [T]$, where $\mathcal{S}_i \subset \mathcal{S}_j$ for $j > i$. [9] An example is 2-phase training, where the first phase uses a subset randomly sampled from the population, and the second phase optimizes on the full population. This is similar to the two-set training proposed in Charton and Kempe (2024), where each batch is a mix of samples from two sets: one small set which is repeated, and one large set consisting of online samples. General multi-phase training requires specifying the sizes and number of steps per phase. A heuristic is to make (1) the first few subsets sufficiently small so that the model can quickly achieve non-trivial performance on the dataset, and (2) the final subset $\mathcal{S}_T$ sufficiently large to ensure good generalization.

## C.2 TRAINING SETUPS

**SIM** We train Transformers with 2 layers and 4 heads and embedding dimension 64 on the SIM task, where the link function is degree 3 Hermite polynomial and dimension $n = \{40, 50\}$. With the fixed compute budget as batch size 128 and number of optimization steps 5000, a simple 2-phase repeat can accelerate training.

**Sparse parity** For Figure 3, each heatmap was created with roughly 42 million training runs done on a single A100 in four hours. Transformer experiments are using encoder-only structure to preserve a permutation-invariant structure of the parity task.

**Remark 5** (Population gradients for sparse parity). Sparse parity has a special structure that the population gradient of the first layer weight reveals information about the support (Barak et al., 2022), hence training with the full population can in theory leverage this information and converge quickly. However, the population gradient signal is exponentially small, hence leveraging the signal requires an exponentially large learning rate (on the order of $d^k$) which is infeasible due to numerical limitations and movements from the second layer. Our experiments confirm this and we did not see strict speedup from increasing the learning rate.

---

[8] We additionally experiment with Gaussian initialization (i.e., $W_{ij} \sim \mathcal{N}(0, 1/\sqrt{d_{\text{in}}})$) whose standard deviation differs from that of the uniform distribution differs by a factor of $\sqrt{3}$. We get the same conclusions; details can be found in Appendix C.

[9] We experimented with an alternative where each subsets are drawn independently without requiring to be a superset of the previous subsets. The results were similar, so we keep the subset requirement which has the additional benefit of smaller sample complexity.

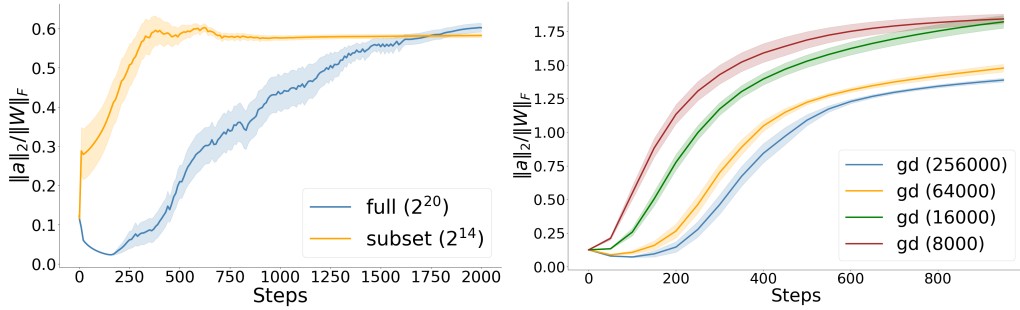

Figure 7: **Layer norm ratio $\|a\|_2/\|W\|_F$ increases.** Results are shown for MLP on (20, 6)-parity and SIM trained with gradient descent.

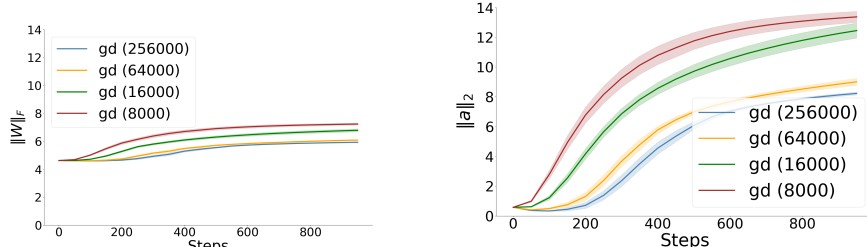

Figure 8: **Layer norm growth during training**. Results are shown for MLP on SIM trained with gradient descent.

**Modular addition** Modular addition runs use a 4-phase training strategy, where the dataset size increases in each phase. Experiments are performed with 2-layer decoder-only Transformers.

**In-context linear regression** We use Transformers 2 layers and 4 heads and embedding dimension 64 on the in-context linear regression task with the number of context examples $k = 15$ and dimension $n = 4$. During training, the loss is computed on the last token.

## C.3 TRANSFORMER EXPERIMENTS

Figure 9 shows Transformer experiments across various tasks. In addition to parity and single-index model (SIM), we consider two additional algorithmic tasks:

- **In-context linear regression**: the input is a sequence $x_1, y_1, x_2, y_2, \ldots, x_k, y_k, x_q$ of length $2k + 1$. Each sequence has an independently sampled $w \sim \mathcal{N}(0, I_n)$, $x_i \sim \mathcal{N}(0, I_n)$, $y_i = w^\top x_i, \forall i \in [k]$ and the label is given by $y := w^\top x_q$.

- $(N, p)$**-modular addition**: the input are two numbers $x, z \sim \text{Unif}([N])$, and the label is given by $y := (x + z) \mod p$ for some prime $p$. For Transformer experiments, $x, z$ are each represented by $\lceil \log_b N \rceil$ digits in base-$b$, and the output logits have size $p$.

## C.4 DETAILS ON RANDOM LABEL EXPERIMENTS

We experiment with both parity and SIM. For parity, we generate the label for the small dataset by sampling $y$ uniformly from $\{-1, +1\}$. For SIM, we sample a random feature vector $w_{\text{random}} \sim \mathcal{N}(0, I)$ and use $w_{\text{random}}$ along with the true link function to label the small dataset.

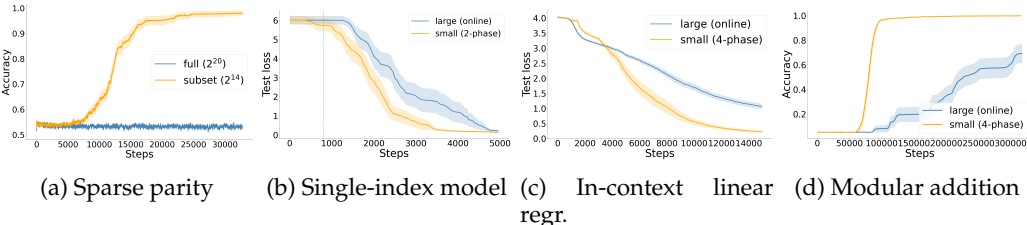

(a) Sparse parity    (b) Single-index model    (c) In-context linear regr.    (d) Modular addition

Figure 9: **Small-vs-large gap exists in various tasks**. Smaller datasets (yellow) lead to faster convergence than bigger datasets (blue). Results are based on 2-layer Transformers.

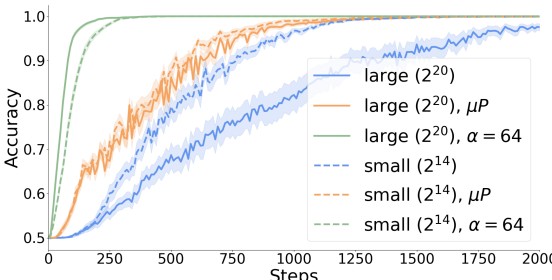

Figure 10: **Comparison to** $\mu P$ (Yang et al., 2022). $\mu P$ and our $\alpha$ scaling both speed up learning compared to the default parameterization, with the optimal $\alpha$ scaling having a slight edge.

## C.5   Additional results on interventions

### C.5.1   Connection to NTK initialization and $\mu P$.

**Optimal scaling and connection alternative parameterizations**   It is worth noting that amplifying contributions from the first layer or balancing different layers in general are classic ideas in the optimization literature (Azulay et al., 2021; Yang et al., 2022; Everett et al., 2024). In particular, the NTK parameterization (Yang and Hu, 2020) is derived to ensure that in at each gradient update, each layer results in the same amount of change in model output. For our 2-layer setup, this implies that for a width-$m$ network, the relative shrinkage on the first layer over the second layer should be proportional to $m^{1/2}$ (Yang et al., 2022). The relative norm adjustment differs from the NTK parameterization.

We empirically identify the best scaling for different model widths. We limit the search of scaling to a single parameter $\alpha$, and adjust the layer scaling by dividing the first layer initialization standard deviation by $\alpha$ and multiplying by $\alpha$ for the second layer. As shown in Figure 11, the optimal $\alpha$ remains constant across model widths, and the learning speedup persists when the scaling far exceeds what NTK parameterization predicts. Related, the $\mu P$ parameterization (Yang and Hu, 2020; Yang et al., 2023), which was developed for maximal updates for feature learning, also leads to different behaviors than layer scaling (Figure 10).

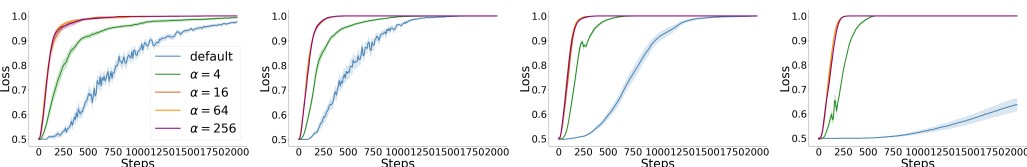

Figure 11: **Initialization scale holds constant across width.** Results are for MLP on (20, 6)-parity trained with (full-batch) GD on $N = 2^{14}$ samples, at various widths $m \in \{32, 64, 256, 1024\}$.

### C.5.2 TRANSFORMER: QK AND MLP SCALING

For Transformers, the effect of scaling is much more subtle due to the use of normalization layers. We thus focus on places where normalization layers are not always represent: the $W_q, W_k$ matrices, and layers of the per-block MLP. The norms of $W_q, W_k$ matrices govern the shape of the softmax attention. Empirically, we find that increasing initialization scale help speed up training; Figure 12 show examples on various tasks. Such scaling connects closely to QK-layernorm, which is known to help mitigate training instabilities and has been adopted as standard practices (Dehghani et al., 2023; Wortsman et al., 2023). Indeed, Figure 12(a) show that removing QK norm slows down learning. The scaling of the per-block MLP acts orthogonally and can be used in conjunction to provide further speedup.

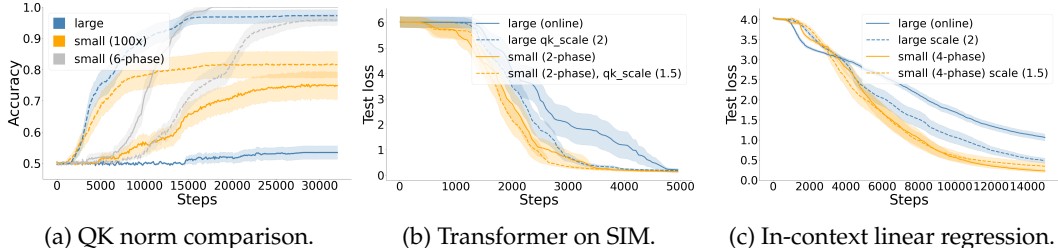

(a) QK norm comparison.    (b) Transformer on SIM.    (c) In-context linear regression.

Figure 12: **QK norm accelerates learning in Transformers.** Results are shown with two-layer Transformers on **(a)** (20,6)-parity, **(b)** SIM, and **(c)** in-context learning regression. Increasing the QK initialization scale can accelerate large-set training. In contrast, the effect of QK-layernorm is more complicated: adding QK-layernorm (dashed lines) speeds up large-set and 100-epoch training, but slows down 6-phase training.

## D IMPLICATIONS

This section discusses implications of our findings.

**Effect of model depth**    Since small-vs-large gap is due to the relative norm growth across layers, this suggests that the gap should be more pronounced when the model depth increases as the scaling effect will percolate exponentially in depth. Our empirical findings confirm this that the small-vs-large gap widens. on both MLP (Figure 14) and Transformers (Figure 15).

**Effect of task complexity**    Increased task complexity appears to increase the gap between large and small dataset training. We see this across tasks and architecture. In Figure 13a, we train an MLP on SIM with link function degree-3 Hermite polynomial and $d = 40, 80$ using GD and the result also shows that the gap grows larger as the task complexity increases. In fig. 13b we compare transformers trained on $(10, 6)$-parity with those trained on $(20, 6)$ parity, and see that the gap grows dramatically.

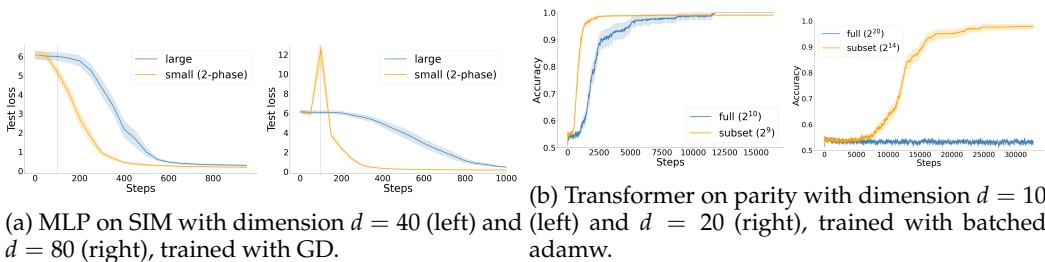

(b) Transformer on parity with dimension $d = 10$

(a) MLP on SIM with dimension $d = 40$ (left) and (left) and $d = 20$ (right), trained with batched $d = 80$ (right), trained with GD.    adamw.

Figure 13: **Increasing the task complexity increases the gap.**

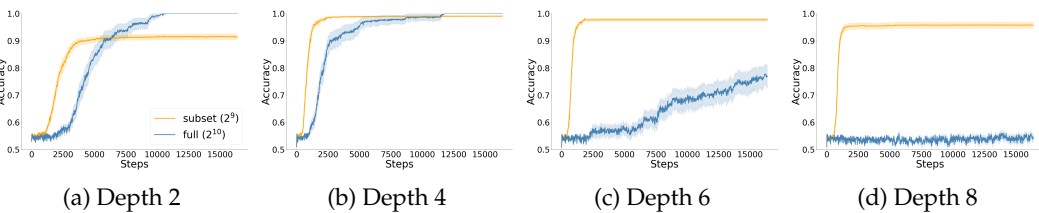

(a) Depth 2      (b) Depth 4      (c) Depth 6      (d) Depth 8

Figure 14: **Increasing depth widens the small-vs-large gap**. Results are from MLP with GD on $(20, 6)$-parity, with width 64 and varying depth.

(a) Depth 2      (b) Depth 4      (c) Depth 6      (d) Depth 8

Figure 15: **Increasing depth widens the small-vs-large gap**. Results are from transformers with batched adamw on $(10, 6)$-parity, with varying number of layers.

**When is data repetition not helpful?** Even though the small-vs-large gap has been observed across tasks, architectures, and optimizers, we do not believe it to exist universally. One classic example is linear regression. While prior work has shown that multi-epoch training improves the statistical complexity for linear regression (Pillaud-Vivien et al., 2018; Lin et al., 2025), no result has suggested an improvement in terms of optimization steps or the compute cost. We hypothesize that the speedup from data repetition is exclusive for non-convex optimization, and a computational-statistical gap is likely required. Connecting to practice, especially the era of large-language model training, we do not expect this phenomena to be directly observable in many real-world language corpora, due to both (near) duplicates widely present in the web datasets, and the lack of clear structure in free-form texts. However, we hypothesize that the small-vs-large gap may be of interest for more structured tasks such as formal reasoning.

