# OpenReview forum: "Less Data, Faster Training: sampling bias from small dataset can speed up training"
_ICLR.cc/2026/Workshop/Sci4DL — Sci4DL 2026_

### Official Review · Reviewer_JHEb · 2026-02-23

**Fit:** 3
**Significance:** 3
**Confidence:** 1

**Summary:**

This paper investigates the small-vs-large gap: under a fixed compute budget, training on fewer unique samples can sometimes yield faster learning and improved performance than training on a larger dataset. The authors argue that common explanations rooted in SGD-specific effects are insufficient, and instead propose a mechanism driven by sampling bias in smaller datasets. They suggest that this sampling bias accelerates relative norm growth across layers, effectively altering layer-wise learning rates and enabling earlier feature-learning. As a result, the gap mainly arises from the effective layer-wise learning rates rather than the size of the dataset; however, a smaller dataset accelerates this.

The paper supports this hypothesis with (i) a theoretical analysis for learning 2-sparse parity using a two-layer MLP, and (ii) empirical evidence through targeted interventions, including random-label training in an initial phase, modified layer initialization scales, and explicit layer-wise learning-rate tuning. Across these experiments, the authors show that the small-vs-large gap can be reduced by controlling the relative layer norms, consistent with their proposed explanation.

**Strengths:**

**1. Significance**: The paper addresses a practically relevant question in modern training regimes, where the prevailing assumption is that more data reliably yields better performance. Identifying conditions under which smaller datasets can accelerate learning, and understanding the mechanism behind this effect, can help researchers make more informed decisions about data collection, compute allocation, and training strategies under resource constraints. Moreover, by demonstrating that the small-vs-large gap persists even under full-batch gradient descent, the authors strengthen the case that explanations based purely on SGD-specific effects are incomplete.

**2. Methodology:** The authors approach the problem both theoretically and experimentally. The experimental design includes multiple interventions (random-label phase, initialization scaling, layer-wise learning-rate tuning) that directly probe the proposed mechanism.

**3. Results:** The experimental results are broadly consistent with the paper’s main claims, (i) that the small-vs-large gap can arise even under full-batch GD and (ii) that the magnitude of the gap is closely tied to differences in relative layer-norm growth, with several interventions reducing or closing the gap in the expected direction.

**Suggestions:**

1. Improve clarity and reporting of experimental details. Several implementation and plotting details are currently missing or ambiguous. For instance, the paper does not clearly describe how test data are sampled and how many test samples are used per evaluation. In addition, some figure legends include numerical values that are not defined anywhere in the text (e.g., Figure 2), and certain visual markers, such as the vertical gray line shown in multiple plots, are not explained. The notion of “two-phase training” is used both in the theorem proof and in the random-label intervention, but it is unclear what “two-phase” corresponds to in Figure 1. Adding explicit descriptions of the evaluation protocol and clarifying all legend values and plot annotations would substantially improve readability and reproducibility.

2. Clarify the connection between the two-phase proof and practical training dynamics. The theoretical analysis relies on a two-phase procedure, but it is not obvious that real training trajectories naturally decompose into the same phases (or analogously switch regimes). It would strengthen the paper to provide additional empirical evidence that standard training exhibits a comparable phase transition (e.g., by identifying consistent change-points in norm ratios, effective layer-wise learning rates, or loss curvature), or to more explicitly frame the two-phase setup as an analytical device and discuss its limitations.

3. Extend experiments beyond synthetic tasks. The two synthetic benchmarks are well-motivated for a mechanistic study, but the paper’s broader claims would be more compelling with at least one or two real-world tasks (or more realistic benchmarks) showing similar behavior. Even a small-scale experiment on a standard dataset would help assess external validity and demonstrate practical relevance.

4. Improve presentation of the task-complexity results in the appendix. The appendix plots that vary task complexity are helpful, but they could be reorganized to make comparisons easier. For example, grouping results for the same task across different complexity levels within a single figure (or as clearly labeled subpanels). Additionally, some figure captions (e.g., Figure 13) appear to be broken.

---

### Official Review · Reviewer_jH7A · 2026-02-26

**Fit:** 2
**Significance:** 2
**Confidence:** 2

**Summary:**

This paper examines whether using a smaller dataset can lead to faster training under a fixed compute budget. There are experiments on synthetic tasks which show that smaller datasets can reach low error with less compute, even under full batch gradient descent. It is also empirically and theoretically suggested that sampling bias changes how layer norms grow during training, which affects learning dynamics and speeds up feature learning

**Strengths:**

The empirical results are cleanly presented and show a consistent effect across several controlled synthetic tasks, which makes the core phenomenon easy to observe and understand

The paper does not rely only on experiments but also proposes a concrete explanation based on layer norm growth and supports it with additional intervention studies, such as learning rate sweeps and scaling experiments

The inclusion of a theoretical section, even in a simplified setting, helps provide intuition for why the effect might occur rather than leaving the findings purely empirical

**Suggestions:**

This paper compares different dataset sizes, but does not clearly state whether hyperparameters were tuned independently per dataset size. Reporting this tuning or performing some sensitivity analyses would strengthen the comparisons. It's mentioned in lines 1001 and 1002 that there is a LR sweep, but the explicit inclusion of your learning rates and other hyperparameters might help strengthen your experiments

Since the proposed mechanism relies on sampling bias from smaller datasets, additional reporting of performance variability across independent subset draws (that is, repeating the experiment with many different randomly selected small datasets and showing how much the results vary) would further strengthen the reproducibility of the claims

The strongest empirical evidence from this paper is on sparse parity and SIM-style synthetic tasks, and extending experiments to larger benchmarks would clarify how broadly the phenomenon applies beyond controlled feature-learning settings, perhaps something past a toy setting, i.e. MNIST

---

### Official Review · Reviewer_w2EF · 2026-02-27

**Fit:** 3
**Significance:** 3
**Confidence:** 2

**Summary:**

The paper studies the phenomemom where (under a fixed compute budget) smaller dataset often converge faster. They posit this is due to the sampling noise affecting the layerwise norm growth. They present empirical evidence which 1. replicates work in literature 2. rules out alternative explanations 3.highlights the layerwise nature of the norm growth. They analyze an minimal model on (primarily) two synthetic scenarios.They also theoretically demonstrate, under certain assumptions, a bound on the number of steps in 2 phase training (sub sampled vs larger dataset).

Overall their paper makes a interesting case and merits further discussion within the workshop.

**Strengths:**

The authors' paper has several strong points:
1. The thoeretical statement (though with significant assumptions) mirrors empirical evidence ad helps generalise conclusions.
2. Several alternative explanations are ruled out e.g. CSQ vs SQ, Var(SGD), Distribution bias
3. The random labels exp. is especially supportive of their core argument.

**Suggestions:**

While the authors present a strong paper. It would be good to see consideration/discussion on the following points:
Conceptually:
1. Elaborating and clarifying the idea of resources budget. They authors mention steps but do not take into account the complexity of steps e.g. different training algorithms. They also do not consider other resources like memory.
2. The situations modelled used are toy models and use significant assumptions about both model, training method, and regimen. It would be good to see a discussion of how this is applicable to actual real deep models. [e.g. without carefully bounded norms, or models with normalisation strategies, or tasks on real (larger more diverse) datasets].
3. The authors argument is highly reliant on initialisation geometry. e.g. dependence on proximity to saddle point. These need better rationalization.

Presentation/structure:
1. Greater discussion on the limitations required - not just in the appendix.

---

### Meta-Review · Area_Chair_EW89 · 2026-03-02

**Recommendation:** Accept

**Metareview:**

The paper studies the role of sampling biases of smaller datasets in driving feature learning. The methodology and results are interesting and well-aligned with the workshop.

---

### Decision · Program_Chairs · 2026-03-02

Accept